# Single-cell and spatial transcriptomics analysis of non-small cell lung cancer

Marco De Zuani [1,2,3,4,11], Haoliang Xue [1,2,3,4,11], Jun Sung Park [1,2,5], Stefan C. Dentro[5,6], Zaira Seferbekova [5], Julien Tessier[7], Sandra Curras-Alonso[8], Angela Hadjipanayis[7], Emmanouil I. Athanasiadis [2,9], Moritz Gerstung[2,5,6], Omer Bayraktar [1,2] & Ana Cvejic [1,2,3,10] ✉

Lung cancer is the second most frequently diagnosed cancer and the leading cause of cancer-related mortality worldwide. Tumour ecosystems feature diverse immune cell types. Myeloid cells, in particular, are prevalent and have a well-established role in promoting the disease. In our study, we profile approximately 900,000 cells from 25 treatment-naive patients with adeno-carcinoma and squamous-cell carcinoma by single-cell and spatial tran-scriptomics. We note an inverse relationship between anti-inflammatory macrophages and NK cells/T cells, and with reduced NK cell cytotoxicity within the tumour. While we observe a similar cell type composition in both adeno-carcinoma and squamous-cell carcinoma, we detect significant differences in the co-expression of various immune checkpoint inhibitors. Moreover, we reveal evidence of a transcriptional "reprogramming" of macrophages in tumours, shifting them towards cholesterol export and adopting a foetal-like transcriptional signature which promotes iron efflux. Our multi-omic resource offers a high-resolution molecular map of tumour-associated macrophages, enhancing our understanding of their role within the tumour microenvironment.

Lung cancer is the second most commonly diagnosed cancer and the first cause of cancer death worldwide[1], with a 5-year survival of ~6% in patients with the most advanced stages[2]. Non-small-cell lung cancer (NSCLC) is the most common type of lung cancer (~85% of total cases), followed by small-cell lung cancer (15% of total cases)[3]. Lung cancer is a complex disease in which the tumour microenvironment plays a cri-tical role and macrophages (Mϕ) are intimately involved in the pro-gression of the disease. In particular, tumour-associated Mϕ (TAMs) can exhibit a dual role, contributing to tumour promotion by suppressing the immune response, facilitating angiogenesis, and aid-ing in tissue remodelling, but also tumour suppression by promoting inflammation and engaging in cytotoxic activity against cancer cells[4,5]. The intricate interplay between lung cancer and Mϕ highlights the importance of understanding their dynamic relationship in order to develop more effective therapeutic strategies.

Within NSCLC, adenocarcinoma (LUAD) is the most common histological subtype, followed by squamous-cell carcinoma (LUSC). Lobectomy (i.e., the anatomical resection of a lung lobe) is currently

[1]Wellcome Sanger Institute, Wellcome Genome Campus, Hinxton, UK. [2]OpenTargets, Wellcome Genome Campus, Hinxton, UK. [3]Department of Haema-tology, University of Cambridge, Cambridge, UK. [4]Wellcome Trust—Medical Research Council Cambridge Stem Cell Institute, Cambridge, UK. [5]European Molecular Biology Laboratory, European Bioinformatics Institute EMBL-EBI, Wellcome Genome Campus, Hinxton, UK. [6]Division of Artificial Intelligence in Oncology, DKFZ, Heidelberg, Germany. [7]Precision Medicine and Computational Biology, Sanofi, Cambridge, MA, USA. [8]Precision Medicine and Computa-tional Biology, Sanofi, Paris, France. [9]Medical Image and Signal Processing Laboratory (MEDISP), Department of Biomedical Engineering, University of West Attica, Athens, Greece. [10]Biotech Research & Innovation Centre (BRIC), University of Copenhagen, Copenhagen, Denmark. [11]These authors contributed equally: Marco De Zuani, Haoliang Xue. ✉e-mail: ana.cvejic@bric.ku.dk

the gold standard for the treatment of early stages of NSCLC (stage I/II), while patients with unresectable stage III or metastatic stage IV NSCLC are treated with a combination of chemotherapy and neoadjuvant targeting vascular endothelial growth factor (VEGF) or immune checkpoint inhibitors (ICIs) like PD1, PDL1 and CTLA4. Advancements made in the last decade in uncovering predictive biomarkers have paved the way for novel therapeutic prospects in the fields of targeted therapy and immunotherapy on the basis of tumour histology and PDL1 expression[6].

A number of studies have employed single-cell technologies to explore transcriptional changes in NSCLC[7–9]. They have extensively examined the lung tumour microenvironment revealing diverse T-cell functions linked to patient prognosis, relevance of diversity of B cells in NSCLC for anti-tumour therapy, multiple states of tumour-infiltrating myeloid cells, proposing them as a new target in immunotherapy, as well as the association of tissue-resident neutrophils with anti-PDL1 therapy failure[7,10–14]. They further unveiled tumour heterogeneity and cellular changes in advanced and metastatic tumours[8,9] as well as tumour therapy-induced transition of cancer cells to a primitive cell state[15]. In many of these studies, a limited number of cells was analysed per patient, and often there was no systematic collection of patient-matched non-tumour tissue, thus restricting dissection of the biological heterogeneity within tumour and adjacent non-tumour tissue. Additionally, with some exceptions[9,14], LUAD and LUSC were considered as a single entity thus hindering the investigation of specific hallmarks of the two cancer types which are radically distinct both at the molecular and pathological level. While single-cell RNA-seq (scRNA-seq) can identify cell types and their states at high resolution within tissues, it lacks the capability to pinpoint their spatial distribution or capture the local cell–cell interactions as well as ligands and receptors that mediate these interactions. Therefore, impeding our ability to fully explore the tumour microenvironment (TME) and the complexity of cell–cell interactions therein.

To overcome above mentioned limitations, we combined scRNA-seq data from nearly 900,000 cells from 25 treatment-naive patients with LUAD or LUSC and spatial transcriptomics from eight patients to investigate the differences in cellular organisation in tumour and adjacent non-tumour tissue. We further examined Mɸ populations and molecular changes they undergo in the tumour environment, some of which resemble those observed in Mɸ during human foetal development.

## Results
### ScRNA-seq and spatial atlas of NSCLC samples
To determine the heterogeneity of immune and non-immune cellular states and their spatial landscape in LUAD and LUSC, we collected lung tissue resections from 25 treatment-naive patients with either LUAD ($n = 13$), LUSC ($n = 8$) or undetermined lung cancer (LC, $n = 4$), and two healthy deceased donors (Fig. 1A, B and Supplementary Data 1). We collected both tumour and matched normal non-tumorigenic tissue (i.e., background), isolated CD45+ immune cells (Supplementary Fig. 1A) as well as tumour and other non-immune populations (using CD235a column to deplete erythroid cells), and performed scRNA-seq. In addition, tumour and background tissue sections from eight patients (of the aforementioned 25) were processed for spatial transcriptomics using the 10x Genomics Visium platform ($n = 36$ sections in total) (Fig. 1A and Supplementary Data 1).

### Tumours exhibit a higher diversity of immune and non-immune cells compared to adjacent lung tissue
Following quality control (QC) on the scRNA-seq dataset, we identified 895,806 high-quality cells in total, of which 503,549 were from tumour and 392,257 from combined background and healthy tissue (from here on referred to as B/H). After performing normalisation and log1p transformation, highly-variable gene selection, dimensionality

reduction, batch correction, and Leiden clustering, cells originating from tumour and B/H were separately annotated into distinct broad cell types and visualised via Uniform Manifold Approximation and Projection (UMAP) (Fig. 1C, Supplementary Fig. 1B, C, and "Methods"). We identified clusters of myeloid cells with transcriptional signatures of monocytes, macrophages, dendritic cells (DCs), as well as mast cells, natural killer (NK) cells, T cells, B cells and non-immune cells (Fig. 1C, D). We did not detect neutrophilic granulocytes, most probably due to their sensitivity to degradation after collection and in particular to the freezing-thawing cycle. Finally, we identified a cluster characterised by the co-expression of myeloid (LYZ, CD68, CD14, MRC1) and epithelial genes (KRT19, EPCAM) (Fig. 1D–F). These cells were found within the tumour and exhibited similarities to previously described cancer-associated macrophage-like cells (CAMLs)[16–18]. CAMLs represent a distinct population of large myeloid cells with concomitant epithelial tumour protein expression[19]. These unique cells have been observed in blood samples of patients with various malignancies, including NSCLC[20]. The abundance of CAMLs exhibits a direct correlation with response to therapeutic interventions, highlighting their functional significance[21]. Even after further subclustering, CAMLs maintained their distinct dual myeloid-epithelial signature (Supplementary Fig. 1D). It is noteworthy that doublet detection software Scrublet assigned a low doublet score to CAMLs, suggesting their expression profile is unlikely to be explained as a combined signature arising from the coincidental sequencing of a tumour cell and a macrophage (Supplementary Fig. 1E). All clusters included cells from multiple patients, with the cluster size ranging from 2520 to 124,459 cells (Supplementary Fig. 1F, G). Furthermore, we conducted reference-query mapping using scArches[22] to confirm the consistency of our annotations in the tumour and B/H dataset (Supplementary Fig. 2A–C and Supplementary Notes).

The composition of the immune and non-immune compartment was markedly different between the tumour and background. In the tumour, we detected fibroblasts and a decrease in the fraction of lymphatic endothelial cells (LECs) ($P_{adj} = 0.0025$, Fig. 1G and Supplementary Data 2). Furthermore, the population of epithelial cells showed higher diversity, with the presence of alveolar type II (AT2), atypical epithelial cells which downregulated epithelial markers (KRT19, EPCAM, CDH1), transitioning epithelial cells which upregulated myeloid markers (LYZ), and cycling epithelial cells in tumour tissues (Fig. 1G, Supplementary Notes, and Supplementary Fig. 2D, E). These differences are in agreement with the fact that in tumour specimens, epithelial cells are likely to be a mixture of mutant tumour and non-mutant normal cells, and suggest that neoplastic transformation leads to further diversity of cell states. We did not detect alveolar type I (AT1) or basal cells, possibly due to their loss during dissociations, as previously reported by others[8].

As previously reported, the proportion of monocytes and immature myeloid cells was significantly reduced in tumour samples compared to background ($P_{adj} = 0.022$ and $P_{adj} = 0.00001$, respectively)[7], while DCs and B cells were overall expanded[7] ($P_{adj} = 0.0023$ and $P_{adj} = 0.0044$, respectively; Fig. 1H and Supplementary Data 3). To get further insight into the cellular composition of tumour versus background tissue, we subclustered each of the broad clusters and identified 46 cell types/states (Supplementary Fig. 2D, E, Supplementary Data 4 and 5, Supplementary Fig. 3, and Supplementary Notes). In the tumour, we found that a significantly higher proportion of NK cells had a lower cytotoxicity phenotype (Supplementary Notes), and that the significant majority of DCs were derived from monocytes (i.e., mo-DC2), (Supplementary Notes) compared to background ($P_{adj} = 0.00002$ and $P_{adj} = 0.00002$, respectively, Fig. 1I and Supplementary Data 6). This is consistent with the monocytic origin of mo-DC2s under inflammatory conditions[23]. Similarly, we found an expansion of B cells expressing LYZ and TNF, and depletion of NKB cells (Fig. 1I and Supplementary Notes). Among T cells, tumour samples showed an accumulation of regulatory

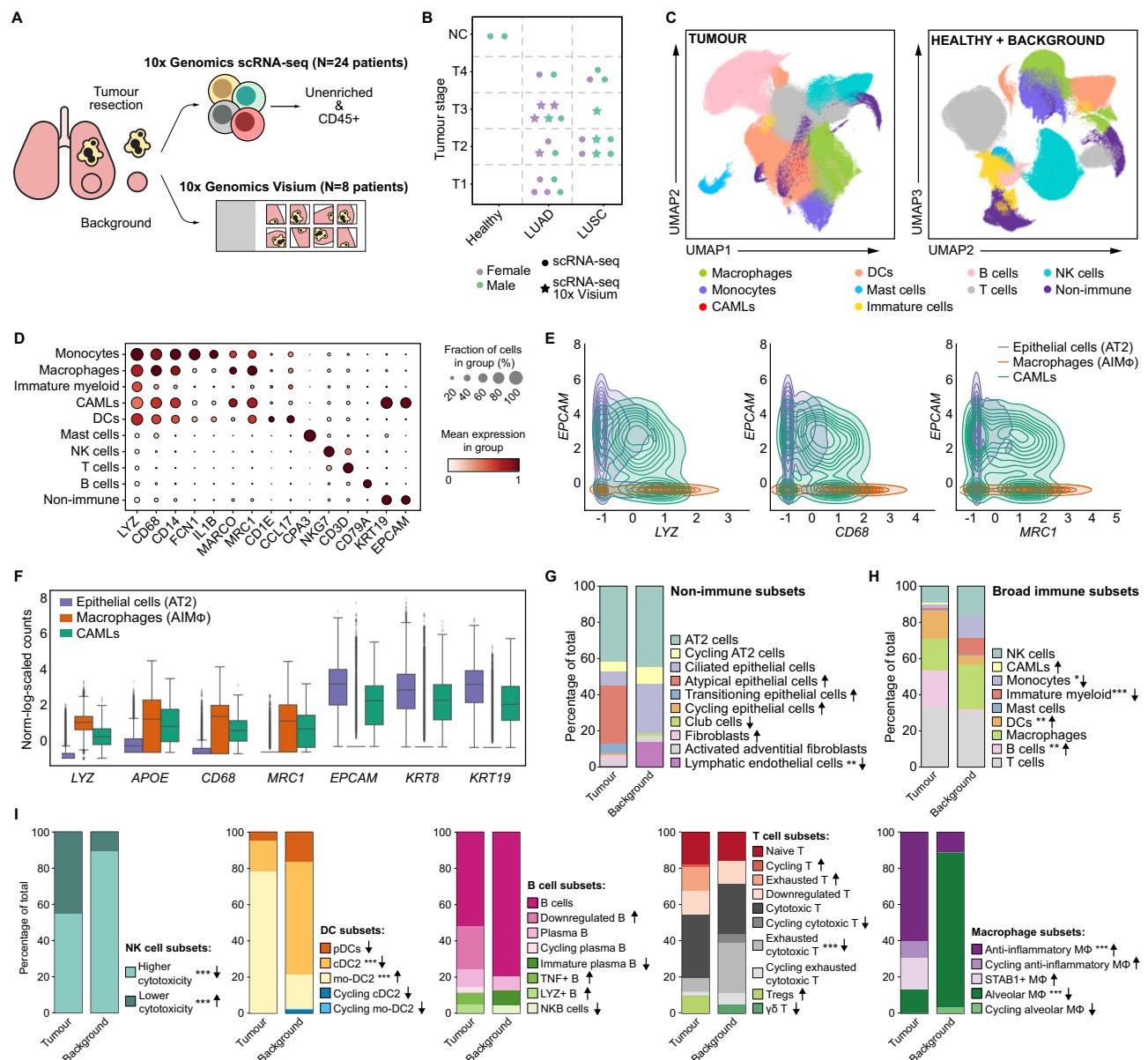

**Fig. 1 | Single-cell transcriptomics reveal the heterogeneity of NSCLC. A** Study overview. Single-cell suspensions of resected tumour tissue, adjacent non-involved tissue (background) and healthy lung from deceased donors were enriched for CD45+ or CD235− and subjected to scRNA-seq. Cryosections of fresh, flash-frozen tumour, background and healthy tissues were used for 10x Visium spatial transcriptomics. **B** Cohort overview. Symbols represent individual patients and performed analyses. **C** UMAP projection of tumour and combined background +healthy datasets. **D** Dotplot of representative genes used for broad cell-type annotations in tumour samples. **E** Contour plot showing the co-expression of myeloid (*LYZ, CD68, MRC1*) and epithelial (*EPCAM*) genes in AT2 cells (44,399 cells), CAMLs (2520 cells) and AIMɸ (16,120 cells). Normalised, scaled and log-transformed gene expression. **F** Boxplot showing normalised, scaled and log-transformed gene expression of myeloid (*LYZ, APOE, CD68, MRC1*) and epithelial (*EPCAM, KRT8, KRT19*) genes in AT2 cells, CAMLs and AIMɸ. Boxes: quartiles. Whiskers: 1.5× interquartile range. **G** Relative proportion of non-immune cell subsets in tumour and background, calculated within the CD235− enrichment. Arrows indicate increase (↑) or decrease (↓) in tumour versus background. Pairwise comparisons by two-sided Wilcoxon rank test and Bonferroni correction for multiple comparisons. **P* < 0.01. Arrows without asterisks indicate that the cell type was found only in tumour or background. **H** Relative proportion of broad immune cells in tumour and background, calculated within all immune cells identified in the CD235- enrichment. Arrows indicate an increase (↑) or decrease (↓) in tumour versus background. Pairwise comparisons by two-sided Wilcoxon rank test and Bonferroni correction for multiple comparisons. *P* < 0.05, **P* < 0.01, ***P* < 0.001. Arrows without asterisks indicate that the cell type was found only in tumour or background. **I** Relative proportion of NK, DC, B, T and macrophage subsets within the broad annotations in tumour and background, calculated within the CD235-enrichment. Arrows indicate increase (↑) or decrease (↓) in tumour versus background. Pairwise comparisons by two-sided Wilcoxon rank test and Bonferroni correction for multiple comparisons. ***P* < 0.001. Arrows without asterisks indicate that the cell type was found only in tumour or background.

T cells (Tregs), known to hinder the immune surveillance of tumours[24] (Fig. 1I). Conversely, there was a reduction of exhausted cytotoxic T cells ($P_{adj}$ = 0.00002) in the tumour and absence of γδ T cells, which have been associated with survival in NSCLC[25] (Fig. 1I and Supplementary Data 6). γδ T cells are capable of recognising and lysing diverse ranges

of cancer cells, and thus have been suggested for a role in pan-cancer immunotherapy[26]. Finally, we saw an increase in heterogeneity and proportion of anti-inflammatory Mɸ (AIMɸ), with a subset of cycling anti-inflammatory Mɸ, *STAB1* + Mɸ (Fig. 1I) and CAMLs (Fig. 1H) being abundantly present in tumour tissue. Interestingly, we found a strong

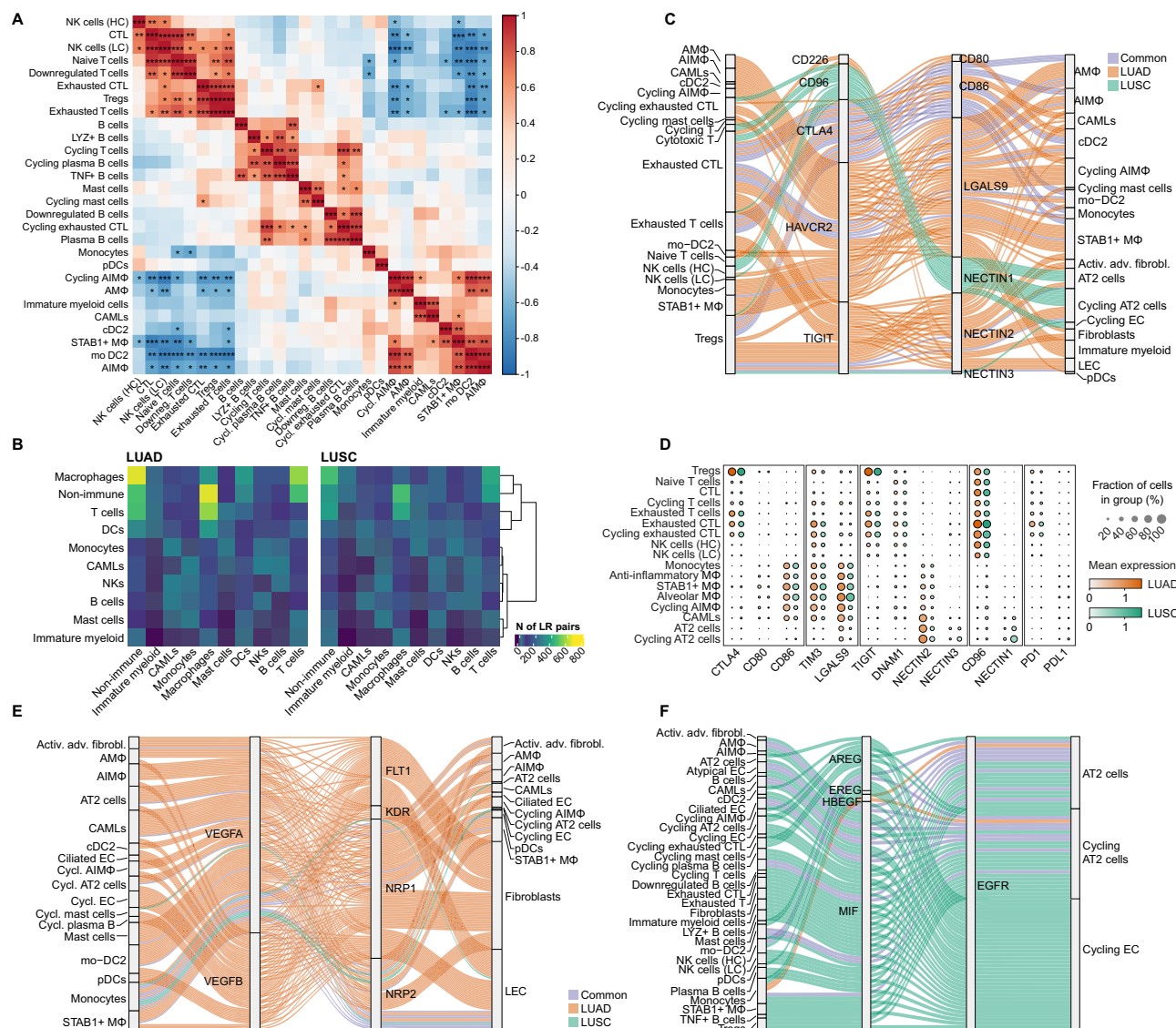

**Fig. 2 | Integrated single cell and spatial transcriptomics uncovers different interaction networks in LUAD and LUSC. A** Heatmap showing the Pearson correlation between the relative cell-type abundance for each immune cell type (calculated within the CD235− enrichment). Colour indicates the Pearson correlation value, asterisks indicate the level of significance of the two-sided association test computed on Pearson's product-moment correlation coefficients (*P < 0.05, **P < 0.01, ***P < 0.001). **B** Heatmap showing the number of LR interactions between all cell types summarised by broad cell annotations in LUAD (left) and LUSC (right). Rows were hierarchically clustered using the complete linkage method on euclidean distances. **C** Sankey diagram showing the tumour-specific interactions in LUAD and LUSC for selected ICIs detected by cellphoneDB. Line colour identifies whether the LR interaction between each cell type was found in LUAD only

(orange), in LUSC only (green) or in both tumour types (blue). **D** Dotplot for the ICI genes and cell types highlighted in (**C**), split by tumour type. The size of each dot represents the percentage of cells in the cluster expressing the gene, while the colour represents the mean normalised scaled log-transformed expression of each gene in each group. **E** Sankey diagram showing the tumour-specific interactions in LUAD and LUSC for VEGFA/B interactors detected by cellphoneDB. Line colour identifies whether the LR interaction between each cell type was found in LUAD only (orange), in LUSC only (green) or in both tumours (blue). **F** Sankey diagram showing the tumour-specific interactions in LUAD and LUSC for EGFR interactors detected by cellphoneDB. Line colour identifies whether the LR interaction between each cell type was found in LUAD only (orange), in LUSC only (green) or in both tumours (blue).

negative correlation between the frequency of *STAB1* + Mφ/AIMφ and T/NK cells across patients, highlighting the key role of Mφ in restraining the infiltration of cytotoxic cells in the lung tumour tissue (Fig. 2A). This is in line with a recent work describing that monocyte-derived Mφ in human NSCLC acquire an immunosuppressive phenotype and restrain the infiltration of NK cells[27].

### LUAD and LUSC have similar cellular composition but utilise different cell−cell interaction networks

LUAD and LUSC have very different prognoses and are often considered as different clinical entities[28]. To examine if differences in

clinical features stem from distinct cellular composition, we compared the frequency of immune and non-immune cell subsets within CD235- samples from LUAD versus LUSC patients. We observed minor differences in cell frequency that did not reach statistical significance after *P* value correction (Supplementary Fig. 4A and Supplementary Data 7 and 8). Furthermore, there was no clear association between the frequency of immune and non-immune cells observed in patients and the cancer subtype, cancer stage or sex (Supplementary Fig. 4B, C), suggesting that the TME composition is rather similar in LUAD and LUSC. While LUAD and LUSC shared similar cellular compositions, the observed clinical distinctions may arise from varying intercellular

interactions. Therefore, we examined whether different cell–cell interaction networks were employed within the TME in LUAD versus LUSC. To this end, we identified a putative list of cell–cell interactions exclusively observed in each tumour type environment by inferring statistically significant ligand–receptor pairs (L–Rs) that were not detected in background or healthy and their corresponding cell types, using CellPhoneDB[29]. Although the two tumour subtypes showed a similar interaction network that mostly involved interactions between non-immune cells, AIMϕ and T cells (Fig. 2B), there were also some notable differences.

First, we identified overall a higher number of L–Rs in the LUAD dataset (Supplementary Fig. 4D and Supplementary Data 9–12), which was not driven by a difference in the number of cells in the LUAD ($n$ = 105,749 cells) vs LUSC ($n$ = 230,066 cells) dataset. Secondly, several pairs of immune checkpoint inhibitors (ICI) and their respective inhibitory molecules were differentially co-expressed in LUAD versus LUSC (Fig. 2C, D). For example, *LGALS9-HAVCR2 (TIM3)*, *NECTIN2-CD226 (DNAM1)* and *NECTIN2/NECTIN3-TIGIT* were frequently identified in LUAD, and the putative ICI *CD96-NECTIN1* was found preferentially in LUSC (Fig. 2C, D). In contrast, *CD80/CD86-CTLA4* and *HLAF-LILRB1/2* were found in both tumour subtypes (Fig. 2C, D). LILRBs (leucocyte Ig-like receptors) are emerging as potential targets for next-generation immunotherapeutics as their blocking can potentiate immune responses[30]. The most commonly used immunotherapies for lung cancer block the interaction between PD1 and PDL1, and recent clinical trials suggested that anti-CTLA4 and anti-PD1 combination therapy improved the survival of patients independent of tumour PD1 expression[31,32]. Within our dataset, we did not observe *PD1-PDL1* interactions in either of the tumour subtypes (Fig. 2C, D). Our initial analysis suggests that other ICIs (such as CTLA4, TIGIT, LILRB1/2 and TIM3) might be promising targets in the treatment of NSCLC.

Of the significant L–Rs detected in both LUAD and LUSC we noted several pairs involved in angiogenic signalling in different populations of myeloid cells such as *VEGFA/B-FLT1*, *VEGFA-KDR* and *VEGFA-NRP1/2*. Although *VEGFA* and *VEGFB* were found to be expressed in both LUAD and LUSC, their receptors were more frequently found in LUAD, especially in fibroblasts (Fig. 2E and Supplementary Fig. 4E). Similarly, we observed significant expression of *EGFR* ligands signalling in AT2 and cycling epithelial cells, such as *EGFR-EREG*, *EGFR-AREG*, *EGFR-HBEGF* and *EGFR-MIF*, although MIF expression was found more frequently in cells from LUSC (Fig. 2F and Supplementary Fig. 4F). Finally, we observed key co-stimulatory signals required to support lymphoid cell activation, such as *CD40-CD40LG*, *CD2-CD58*, *CD28-CD86*, *CCL21-CCR7*, and *TNFRSF13B/C-TNFSF13B* (*TACI/BAFFR-BAFF*) (Supplementary Fig. 4G), which are often associated with the presence of ectopic lymphoid organs mainly consisting of B cells, T cells, and DCs i.e., tertiary lymphoid structures (TLS). TLS are usually correlated with the longer relapse-free survival in NSCLC[33].

## Integration of scRNA-seq and spatial transcriptomics validates L–R interactions in situ

The significant L–Rs and their interacting cell types were calculated based on the co-expression of genes in different cell-type clusters from the scRNA-seq dataset using CellPhoneDB. However, in order to discern biologically significant interactions, it is essential to ascertain whether the cell types identified as interacting are indeed physically co-located. To achieve this, we considered how the scRNA-seq-identified cell types are spatially arranged on tissue sections. We applied an integrative approach which combines the scRNA-seq of the tumour and background samples with the spatial transcriptomic (STx) profile of the fresh frozen tumour and background tissue sections. We performed 10× Visium on two consecutive, 10-μm sections, from eight patients, seven of which matched the samples used for the scRNA-seq. We analysed 36 sections in total ($n_{tumour}$ = 20, $n_{background}$ = 16) with an average UMI count of 6894/spot in tumour and 3350/spot in the background. Next, we used cell2location[34] and cell-type specific expression profiles from our scRNA-seq dataset to deconvolute cell-type abundances on the tissue (Fig. 3A, see "Methods").

Once the cell types were resolved on the tissue sections, we examined the frequency of different cell types across all sections from tumour and background tissue. The cell-type abundance in tumour and background were computed by summing up the posterior 5% quantile (q05) value of estimated cell abundance by cell2location, across spots that passed QC ("Methods"). Our analysis confirmed that the differences in the frequency of cell types across all sections in tumour versus background was in line with the results obtained in the scRNA-seq data (Fig. 3B). For example, in tumours we found an increase in the proportion of B cells ($P_{adj}$ = 0.0372) and cycling AT2 cells ($P_{adj}$ = 0.0147) compared to the background tissue, and a decrease in the proportion of immature cells ($P_{adj}$ = 0.0012), NK cells ($P_{adj}$ = 0.0012), and LECs ($P_{adj}$ = 0.00077, Supplementary Data 13 and 14). However, the proportions of other cell types estimated from the scRNA-seq data or the STx data within the tumour or background showed some discrepancies (Supplementary Fig. 4H, I). This was particularly evident within the non-immune populations, where STx estimated higher proportions of LECs, activated adventitial fibroblasts and cycling subsets, compared to scRNA-seq. Disparities in cell proportions between different methodologies were previously shown by others[35,36], underscoring the potential influence of distinct sampling biases inherent to scRNA-seq and STx techniques like Visium. In the case of scRNA-seq, variations in cell digestion sensitivity can lead to differential representation of cell types. Meanwhile, with Visium, discrepancies might arise from variations in the location of tumour resections as well as differences in sample sizes compared to scRNA-seq studies. Nevertheless, the overall concordance in the results obtained by scRNA-seq and Visium suggests that our spatial "map" of different cell types faithfully represents their distribution in the tissue.

Next, we examined the spatial co-localisation of the L–Rs identified by cellphoneDB. The L–Rs were considered to co-localise if both genes were expressed in the same spot and above median value for the given genes across the section spots. We then compared the frequency of spots in which L–R genes were colocalising versus non-colocalising in the matched tumour versus background sections, using a $\chi^2$ test ("Methods"). Due to the low number of tissue blocks collected from LUSC and LUAD patients ($N_{LUSC}$ = 3, $N_{LUAD}$ = 5), the statistical power was not sufficient to perform a comparative analysis between spatial localisation of LUAD/LUSC-specific L–Rs. Nevertheless, we confirmed that several of the aforementioned tumour-specific L–Rs colocalized significantly more in tumour than in background sections, including *NRP1-VEGFA* and the ICIs *NECTIN2-TIGIT*, *LGALS9-HAVCR2*, and *CD96-NECTIN1* (Fig. 3C–E and Supplementary Data 15). Consistent with the cellphoneDB results, we found no significant colocalization of *PD1-PDL1* in the tumour sections.

## CAMLs share similar copy number aberrations (CNAs) with tumour cells

Tumour samples obtained from surgical resection contain both malignant and residual normal epithelial cells. A significant challenge in scRNA-seq of human tumours lies in the differentiation of cancer cells from non-malignant counterparts. Therefore, we applied Copy-number Karyotyping of Tumors (CopyKAT[37]) to discern genome-wide aneuploidy within individual cells. The principle driving the computation of DNA copy number events from scRNA-seq data is rooted in the notion that the expression levels of neighbouring genes can provide valuable information to infer genomic copy numbers within that specific genomic segment. Since aneuploidy is common in human cancers, cells with genome-wide CNAs are considered as tumour cells.

Analysis using CopyKAT revealed extensive, patient-specific CNAs in tumour tissue (Fig. 4A and Supplementary Fig. 5A) but not in the

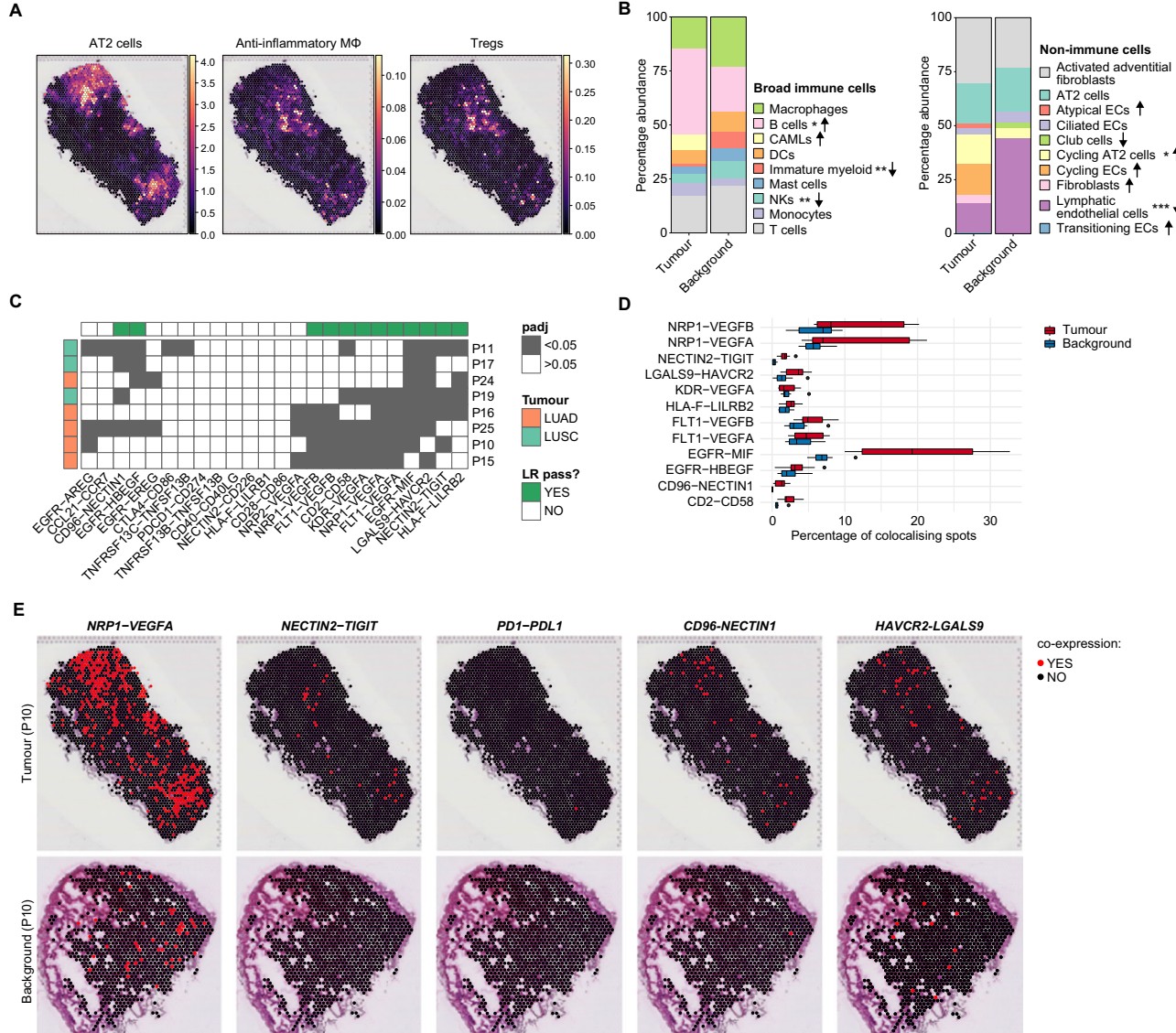

**Fig. 3 | 10x Visium confirms the spatial colocalization of key ligand–receptor pairs. A** Spatial images depicting the cell abundance estimated by cell2location for AT2 cells, AIMϕ and Tregs on a representative tumour section. **B** Relative proportion of immune (left) and non-immune (right) cell types calculated on the cell abundance estimations by cell2location in tumour and background sections. Immune cells were grouped according to their broad annotations. Arrows indicate an increase (↑) or a decrease (↓) in the tumour, compared to the background. Pairwise comparisons were performed with a two-sided Wilcoxon rank test and Bonferroni correction for multiple comparisons. *$P < 0.05$, **$P < 0.01$, ***$P < 0.001$. Arrows without asterisks indicate that the cell type was found only in the tumour or background. Please refer to Supplementary Data 13 and 14 for the exact $P$ values. **C** Heatmap of spatial LR colocalization. LR gene pair co-expression was estimated in each spot for all sections, and the frequency of colocalising vs.

non-colocalising spots in the tumour and background was compared using a $\chi^2$ test followed by Bonferroni multiple comparison correction. Dark-grey tiles indicate that the frequency of colocalising gene pairs was significantly different in tumour and background sections. Green column annotations indicate the LR pairs which were significant in at least four out of eight patients. Row annotations indicate tumour type. **D** Boxplot showing the frequency of colocalising LR pairs significantly different in tumour vs background in each section analysed. $N = 8$ patients. Boxes are plotted with default settings in the Python Seaborn package, i.e., boxes show quartiles with whisker length being 1.5 times the interquartile range. Source data is provided as a Source Data file. **E** Spatial images depicting the location of spots in which the LR pair was found co-expressed in tumour (top) and background (bottom), for *NRP1-VEGFA, NECTIN2-TIGIT, PD1-PDL1, CD96-NECTIN1* and *HAVCR2-LGALS9*. Representative sections from one patient.

background. Within individual tumour samples, the CNAs were detected in AT2 and cycling AT2 cells, and in some patients these genetic alterations were shared between AT2/cycling AT2 cells and atypical epithelial cells, suggesting a close lineage relationship between different epithelial subpopulations (Fig. 4A and Supplementary Fig. 5A). We confirmed this finding by inferring the trajectory of non-blood cell populations in tumour using Partition-Based Graph Abstraction (PAGA)[38]. PAGA showed differentiation continuity between AT2 cells, cycling AT2/epithelial cells, and atypical epithelial cells on one side and ciliated epithelial cells and transitioning epithelial

cells on the other (Fig. 4B). Furthermore, blinded histological evaluation confirmed the overlap between pathologist-defined tumour sites and AT2 and cycling AT2 cells predicted by cell2location, suggesting their tumour cells status (Fig. 4C). Less overlap was observed for atypical epithelial cells (Fig. 4C). The differential expression analysis (DEA) of AT2 cells from tumours compared to background showed upregulation of genes involved in hypoxia, TP53 pathways, and metabolic rewiring in tumours. AT2 cells in tumour-upregulated genes involved both in glycolysis and oxidative phosphorylation (Fig. 4D and Supplementary Data 16). While the importance of glycolysis in tumour

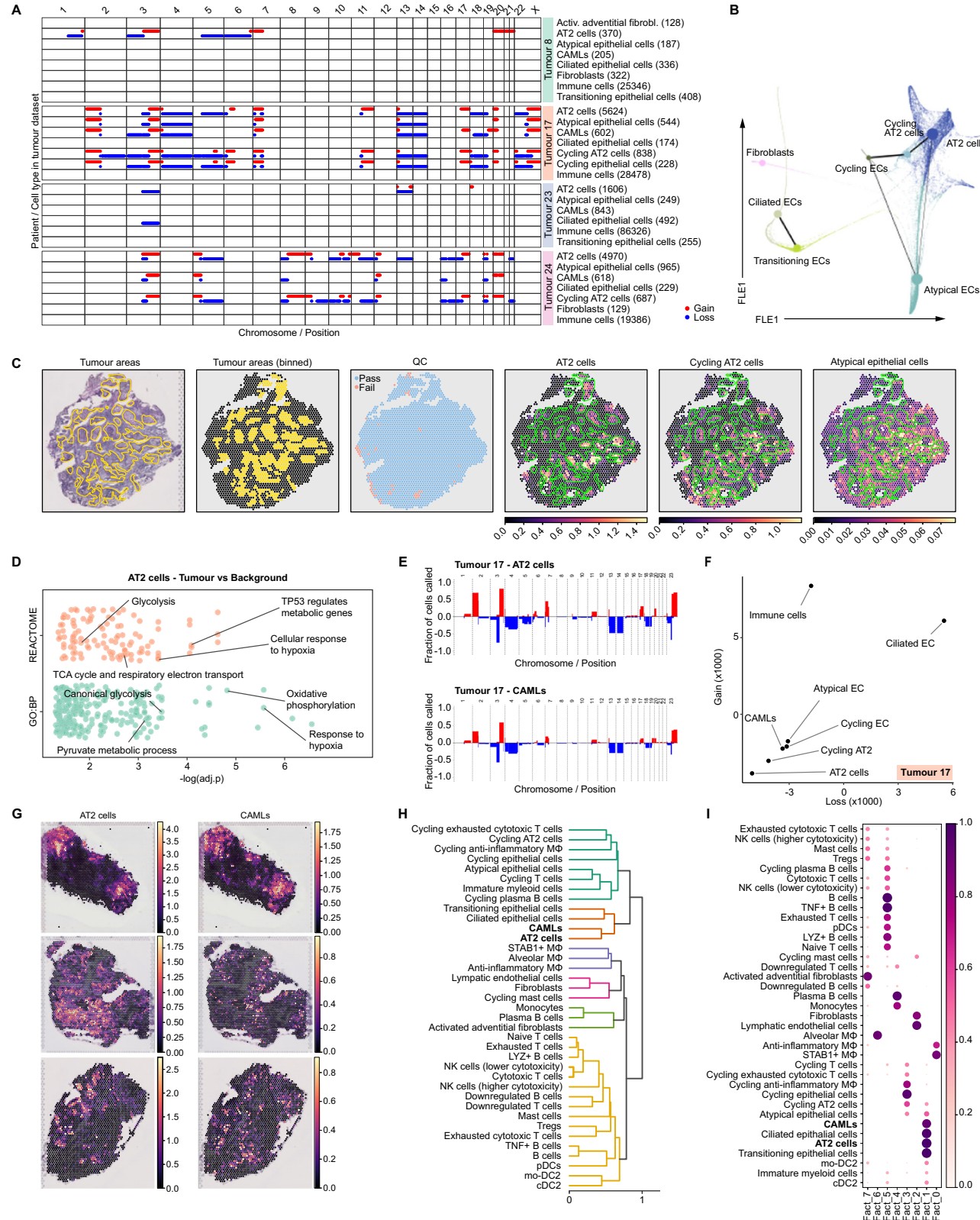

cells is well-established[39], it was recently reported that human NSCLC use glucose and lactate to fuel the tricarboxylic acid (TCA) cycle[40]. In addition, the tumour AT2 cells were noted to express more *LYPD3* compared to background AT2 cells (log2FC = 2.04, $P_{adj}$ = 0.039, Supplementary Data 16), an adhesion protein which has previously been connected to poor prognosis in NSCLC and is currently being targeted in preclinical and clinical studies[41,42].

Interestingly, the population of CAMLs also showed substantial CNAs that were similar to those of AT2 cells and cycling AT2 cells from the same patient (Fig. 4A, E and Supplementary Fig. 5A, B). To measure the difference of the distribution of genomic gain and loss between cell types in a statistically robust manner, we calculated the Kullback–Leibler (KL) divergence (Fig. 4F and Supplementary Fig. 5C). CAMLs had KL divergence values comparable to CNA-harbouring

**Fig. 4 | CAMLs share tumour CNAs and colocalise with tumour cells. A** CNA analysis. The plot shows chromosomal gains (red lines) and losses (blue lines) estimated by CopyKat in each chromosome arm for different cell types and patients in the tumour dataset. All immune cell types were grouped together for plotting purposes. **B** PAGA graph overlaid on the diffusion maps (force-directed layout—FLE embedding) computed for non-immune cell types in tumour. **C** First three panels— Representative blind annotations from a qualified pathologist, indicating the areas of tumour infiltration (left), binning of the tumour area on the Visium spots (centre) and the spots that passed QC (right). The last three panels—cell2location estimation for AT2 cells (left), Cycling AT2 cells (centre) and Atypical epithelial cells (right) on the same sections, overlaid with the pathologist's annotation for the tumour infiltration (green contour). **D** Overrepresentation analysis on gene ontology—biological processes (GO:BP) and REACTOME database by clusterProfiler R package, using DEGs upregulated by AT2 cells in tumour vs background. Source data is provided as a Source Data file. **E** Detailed overview of CNAs in AT2 and CAMLs from the tumour of one representative patient. Bars indicate the frequency of cells harbouring chromosomal gains (red bar) or losses (blue bars) in specific chromosomal regions. **F** Scatterplot of the KL divergence for losses (x axis) and gains (y axis) between each cell type in the tumour dataset calculated using their gain and loss distribution. All immune cell types were grouped together for plotting purposes. **G** Spatial images depicting the cell abundance estimated by cell2location for AT2 cells and CAMLs on three representative tumour sections. **H** Hierarchical clustering of the correlation distance calculated on cell-type composition (as estimated by cell2location) across spots that passed QC in all tumour sections. **I** Non-negative matrix factorisation built on the q05 estimation of cell-type abundance across spots that passed QC (as estimated by cell2location) in all tumour sections.

tumour cells, thus confirming the similarity of their CNA profiles (Fig. 4F and Supplementary Fig. 5C). As CAMLs co-expressed a wide array of myeloid genes as well as typical epithelial genes (Fig. 1D–F and Supplementary Fig. 1D), had a low doublet score and shared the same CNA signature as tumour cells, we hypothesised that these cells might represent a subset of Mφ tightly attached to a cancer cell. It is possible that these Mφ were undergoing phagocytosis or fusion.

CAMLs have been previously isolated from peripheral blood of cancer patients and described to facilitate circulating tumour cells seeding of distant metastases[16]. Our analysis suggested that CAMLs can also be isolated from tumour tissue. To validate that CAMLs are in physical proximity to tumour cells in situ we examined our STx sections. We calculated across all sections (8 patients, $n_{sections} = 20$) the Pearson correlation between the relative abundance of the cell types that reside in the same spot and are therefore co-localised. Our analysis showed that CAMLs indeed co-localised with AT2 cells (Fig. 4G, H). We confirmed this finding using non-negative matrix factorisation (NMF) on the absolute cell-type abundances estimated by cell2location that defined factors of co-occurring cell states (Fig. 4I).

To determine the specific Mφ population from which CAMLs likely originate, we employed PAGA to elucidate the differentiation path of the myeloid cell population in our tumour dataset (Supplementary Fig. 5D). The analysis revealed continuity of the differentiation transitions between diverse populations of myeloid cells[43]. Within the PAGA trajectory, alveolar Mφ (AMφ) and AIMφ showed high PAGA connectivity indicating their high transcriptional similarity. Both AIMφ and AMφ showed the strongest connectivity on the PAGA trajectory with STAB1 + Mφ which, in turn, were linked with CAMLs. In line with trajectory analysis, CAMLs co-expressed many of the genes specific to STAB1 + Mφ (Supplementary Fig. 2A), supporting the hypothesis that CAMLs are likely derived from STAB1 + Mφ following their close interaction with tumour cells. Finally, DEA analysis between CAMLs from LUSC *versus* LUAD patients, showed upregulation of *KRT17*, *KRT5* and *KRT6A* in LUSC samples (Supplementary Data 17). These *KRT* genes were previously identified as markers of LUSC in multiple studies[44,45], which supports hypothesis that CAMLs arise from the interaction between Mφ and tumour cell.

## TAMs promote cholesterol and iron efflux in tumour

Mφ, traditionally categorised into distinct M1 (classically activated) and M2 (alternatively activated) phenotypes, are now understood to exist along a dynamic spectrum of functional states[46]. This concept of Mφ plasticity underscores their ability to seamlessly transition between pro-inflammatory and anti-inflammatory roles in response to intricate cues from their microenvironment (Supplementary Fig. 5D). To better understand the transcriptional changes that different Mφ populations undergo in the TME, we performed DEA. In tumours, both AMφ and AIMφ upregulated genes involved in cholesterol and lipid transport and metabolism (such as *ABCA1, APOC1, APOE, FABP3* and *FABP5*) compared to the background tissue (Fig. 5A, B and Supplementary Data 18 and 19). Cholesterol plays a vital role in tumour

growth due to the high demand of newly synthesised cellular membranes during cancer cell proliferation. Hypoxia-related genes were upregulated in AT2 cells in tumour compared to the background (Fig. 4D), which can promote cholesterol auxotrophy in tumour cells by suppressing cholesterol synthesis, thereby forcing them to rely on exogenous cholesterol uptake[47]. In our dataset, we detected higher expression of the cholesterol exporter *ABCA1* and no expression of low-density lipoprotein receptor (*LDLR*) in AMφ and AIMφ, the latter gene being responsible for the uptake of cholesterol-carrying lipoprotein particles into cells, suggesting preferential export of cholesterol from TAMs to the TME (Fig. 5A). Interestingly, we also noted a high expression of *TREM2* in both AMφ and AIMφ (Fig. 5A), which plays a prominent role in efflux of cholesterol in microglia[48–50]. To validate the increased levels of cholesterol in the TME, we stained matched tumour and background tissue sections with BODIPY™ 493/503, a stain targeting cholesterol and other neutral lipids. We found a significant increase in the BODIPY signal in the tumour sections, compared to background tissue (Fig. 5C, D), confirming an increased availability of neutral lipids in the tumour, possibly as a result of an increased export by TAMs.

STAB1 + Mφ were identified in the tumour resections (Fig. 5E–H, Supplementary Fig. 2 and Supplementary Notes), so we used DEA to identify a set of genes that were specific for STAB1 + Mφ compared to tumour AIMφ or AMφ. We identified 20 genes, from here on referred to as "STAB1 signature genes" (Fig. 5I). Interestingly, STAB1 + Mφ uniquely expressed *SLC40A1*, which encodes for the ferroportin, the only known protein that exports ferrous iron from the cytoplasm across the plasma membrane and is key for the iron-releasing activity of macrophages (Fig. 5I, J and Supplementary Data 20 and 21)[51]. Ferroportin-mediated release of free iron by M2 Mφ was reported to promote the proliferation of renal carcinoma cells in vitro, possibly by supporting the high iron requirement due to increased DNA synthesis[52]. Furthermore, compared to AMφ, STAB1 + Mφ expressed lower levels of ferritin heavy chain 1 (*FTH1*) and ferritin light chain (*FTL*) encoding for the iron storer ferritin (Fig. 5J and Supplementary Data 20). Consistent with the hypothesis of their sustained export of free iron to the extracellular milieu, STAB1 + Mφ downregulated genes involved in iron sequestration (Fig. 5K). Taken together, our analysis suggests that macrophages undergo "reprogramming" within the TME and adopt a transcriptional signature that facilitates cholesterol efflux and iron export, thus supporting tumour progression.

## STAB1 + Mφ in tumour tissue undergo oncofoetal reprogramming

Embryonic development shares many characteristics with tumour tissue, including rapid cell division, cellular flexibility, and a highly vascular microenvironment. It has been recently reported that during tumorigenesis, Mφ can undergo oncofoetal reprogramming[53] and acquire a foetal-like transcriptional identity that supports tumour growth and metastasis[53]. Considering that some of the STAB1 signature genes are typically expressed by foetal Mφ (such as *STAB1, FOLR2,*

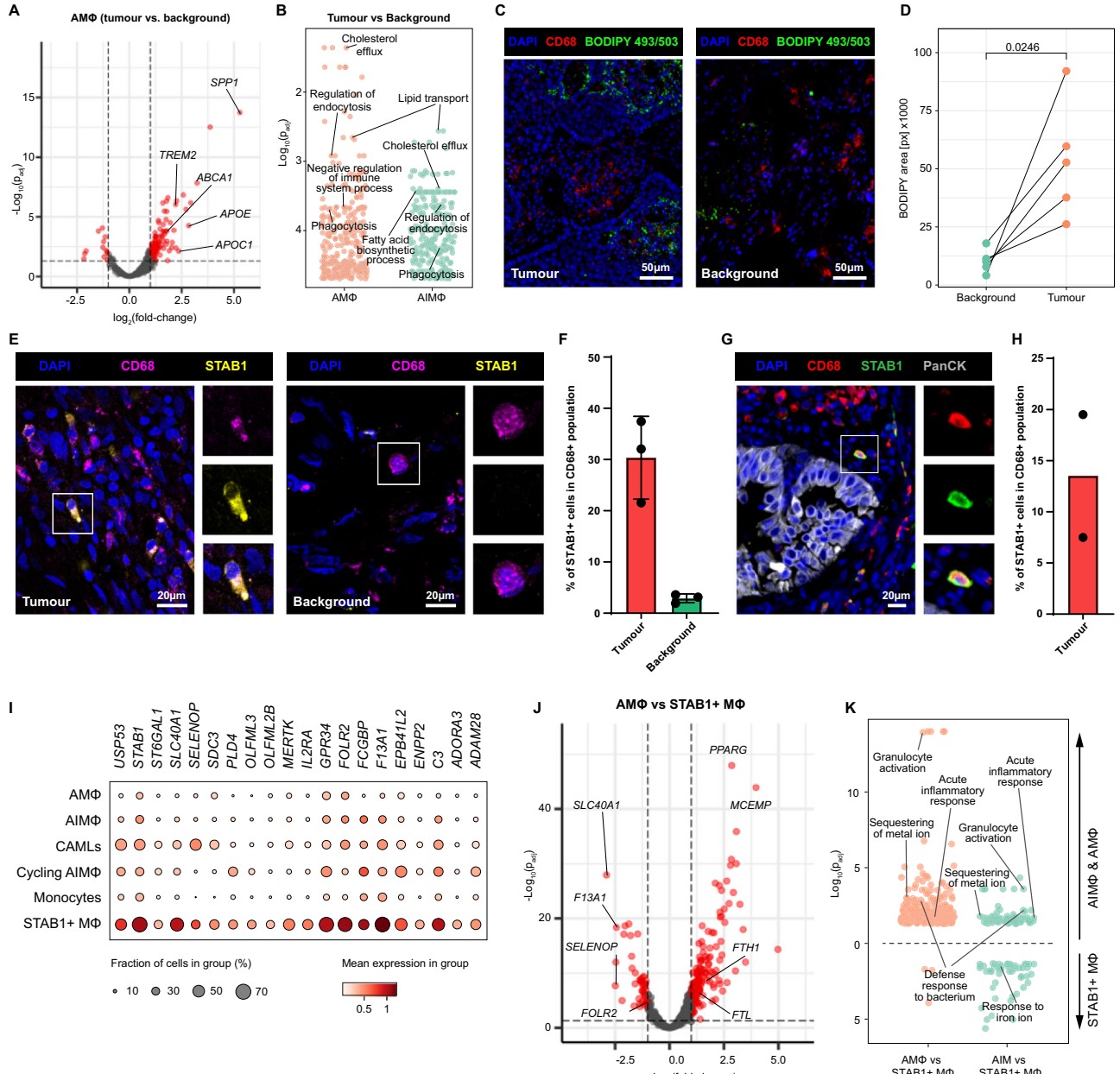

**Fig. 5 | Tumour macrophages undergo oncofoetal reprogramming. A** Volcano plot of DEGs (red) for AIMφ in tumour vs background, extracted using the py_DE-Seq2 package. **B** Overrepresentation analysis on gene ontology−biological processes database by clusterProfiler R package, using the DEGs upregulated by Alveolar Mφ and AIMφ in tumour vs background. Source data is provided as a Source Data file. **C** IHC for CD68 and neutral lipids (BODIPY 493/503) on tumour and background tissue sections. Maximum intensity projection of Z-stacks. Scale bar 50 μm. **D** Area covered by the BODIPY signal in tumour and background section. The difference in BODIPY area coverage was determined with a paired, two-sided t test, matching tumour and background sections from the same patients. *N* = 5 patients. Source data is provided as a Source Data file. **E** IHC for CD68 and STAB1 on tumour (left) and background (right) tissue sections. Maximum intensity projection of Z-stacks. Inlets show a detailed magnification on a single cell. Scale bar 20 μm. **F** Quantification of STAB1+ cells within the CD68+ macrophage population. The fraction of the STAB1 + CD68+ area is shown as a percentage of the total CD68+

area. Data are presented as mean value and standard deviation (*n* = 3 biological replicates). Source data is provided as a Source Data file. **G** Staining for CD68, STAB1 and PanCK on tumour tissue sections. Maximum intensity projection of Z-stacks. Inlets show a detailed magnification on a single cell. Scale bar 20 μm. **H** Quantification of STAB1 + CD68+ cells within the CD68+ macrophage population in NSCLC. Data are presented as mean value and individual data points (*n* = 2 biological replicates). Source data is provided as a Source Data file. **I** Dotplot showing the expression of the "STAB1 signature genes" across all macrophage subsets and CAMLs in tumour. **J** Volcano plot of DEGs identified by py_DESeq2 (red) for Alveolar Mφ vs STAB1 Mφ in tumour. **K** Overrepresentation analysis on gene ontology−biological processes database by clusterProfiler R package, using the DEGs from Alveolar Mφ vs STAB1 Mφ (top) and AIMφ vs STAB1 Mφ (bottom) in tumour (left−upregulated by STAB1 Mφ; right−upregulated by Alveolar Mφ or AIMφ). Source data is provided as a Source Data file.

*SLC40A1, MERTK, GPR34* and *F13A1*)[54], we wanted to explore if further transcriptional commonalities exist between tumour-originating *STAB1* + Mφ and Mφ isolated from human foetal lung. To this end, we combined tumour- and background-originating myeloid cells from our dataset (*n* = 347,364 cells) with myeloid and progenitor cells from a

publicly available foetal lung scRNA-seq dataset[55] (*n* = 6,947 cells) using Harmony. Next, we performed Leiden clustering on the neighbourhood graph and examined how cell types are distributed within the clusters (Supplementary Fig. 6A, B). To examine similarity in their gene expression profile, we applied hierarchical clustering and built a

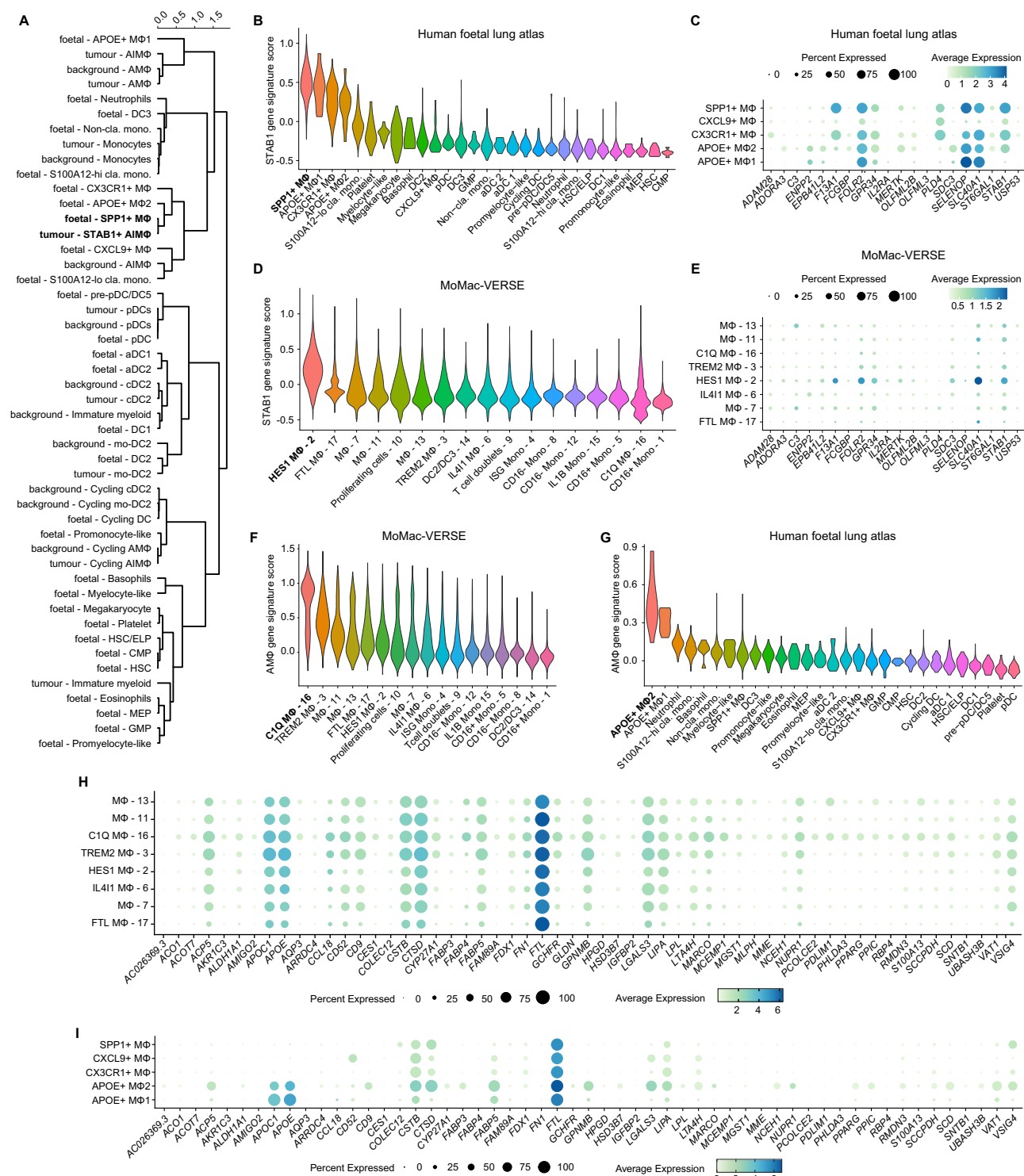

dendrogram by estimating the correlation distance between cell types on the harmonised PC embedding space, under the complete linkage criterion of hierarchical clustering (Fig. 6A).

We observed that tumour cDC2 exhibited the strongest correlation with background cDC2, whereas tumour mo-DC2 displayed the highest correlation with foetal DC2 and, in a broader context, with background mo-DC2. The population of pDC from tumour, background and foetal lung were closely correlated. Similarly, tumour monocytes were correlated with foetal classical monocytes and background monocytes. In contrast, macrophage populations in tumour, and in particular *STAB1* + Mφ, were correlated with foetal

macrophages. *STAB1* + Mφ clustered predominantly with foetal *SPP1* + Mφ (Fig. 6A), which accounted for over 80% of all foetal lung macrophages reported in ref. 55. Consistent with this finding, *SPP1* + Mφ had a high expression of the "STAB1 signature genes" compared to other haematopoietic populations (Fig. 6B, C). Our analysis substantiates the idea that monocytes within the tumour environment, as they undergo differentiation into anti-inflammatory macrophages, acquire a transcriptional signature akin to that of foetal macrophages. This distinctive transcriptional signature was not observed in the macrophages from surrounding normal tissue.

**Fig. 6 | *STAB1* + Mϕ undergo oncofoetal reprogramming. A** Hierarchical clustering of the correlation distance calculated on each cell in the harmonised (tumour myeloid + background myeloid + foetal lung myeloid) PC space. **B** Violin plot showing the expression level of the "STAB1 gene signature" across myeloid cell and progenitor populations identified in a publicly available human foetal lung atlas. **C** Dotplot of the expression of each gene in the "STAB1 gene signature" in selected foetal lung macrophage populations. The size of each dot represents the percentage of cells in the cluster expressing the gene, while the colour represents the mean expression of each gene in each cluster. **D** Violin plot showing the expression level of the "STAB1 gene signature" across the clusters identified in the publicly available MoMac-VERSE dataset. **E** Dotplot of the expression of each gene in the "STAB1 gene signature" in selected macrophage populations from the MoMac-VERSE. The size of each dot represents the percentage of cells in the cluster expressing the gene, while the colour represents the mean expression of each gene in each cluster. **F** Violin plot showing the expression level of the "AM ϕ gene signature" across myeloid cell and progenitor populations identified in the publicly available "MoMac-VERSE" dataset. **G** Violin plot showing the expression level of the "AMϕ gene signature" across myeloid cell and progenitor populations identified in a publicly available human foetal lung atlas. **H** Dotplot of the expression of each gene in the "AMϕ gene signature" in selected macrophages populations identified in the "MoMac-VERSE" dataset. The size of each dot represents the percentage of cells in the cluster expressing the gene, while the colour represents the mean expression of each gene in each cluster. **I** Dotplot of the expression of each gene in the "AMϕ gene signature" in selected foetal lung macrophage populations. The size of each dot represents the percentage of cells in the cluster expressing the gene, while the colour represents the mean expression of each gene in each cluster.

To further examine the prevalence of *STAB1* + Mϕ in other pathologies, including other cancers, we examined the expression of "STAB1 signature genes" across a diverse group of myeloid cells using a published atlas of human monocytes and Mϕ collected from 12 different healthy and pathologic tissues ($n = 140,327$ cells), called MoMac-VERSE[56]. The cluster of "*HES1*+ macrophages" identified in MoMac-VERSE showed the highest expression of the "STAB1 signature genes" (Fig. 6D, E). Similar to *STAB1* + Mϕ, *HES1*+ macrophages accumulated in tumours of lung cancer patients but also liver cancer patients[57] and were suggested to represent a cluster of "long-term resident-like" Mϕ with foetal-like transcriptional signature[56]. In contrast, "C1Q" Mϕ from MoMac-VERSE, which have been described as lung alveolar Mϕ, had a high expression of genes unique to our tumour alveolar AMϕ (from here on referred as "AMϕ signature genes", Fig. 6F, H). In the context of foetal lung, a rare population of APOE + Mϕ, which accounted for less than 1% of all foetal lung macrophages reported in ref. 55, had a high AMϕ signature genes score (Supplementary Notes and Fig. 6G, I, see "Methods").

Taken together, our analysis suggests that tumour macrophages, especially *STAB1* + Mϕ, exhibited a transcriptional signature reminiscent of Mϕ during foetal lung development, suggesting that they have undergone oncofoetal reprogramming within the NSCLC tumour environment.

## Discussion

Our study represents a large single-cell multiomics analysis of samples collected from treatment-naive patients with NSCLC. We integrated scRNA-seq data from nearly 900,000 cells from tumour resections and adjacent non-malignant tissue from 25 treatment- naive patients with spatial transcriptomics to build an atlas of immune and non-immune compartments in lung cancer.

LUAD and LUSC, the two most common NSCLC subtypes, exhibit markedly different prognostic outcomes and have shown potential for subtype-specific therapies[28]. Despite similar cell-type composition, we observed significant differences in the co-expression of several ICIs and inhibitory molecules between LUAD and LUSC, highlighting therapeutic opportunities. LUAD samples frequently expressed *TIGIT* and *TIM3 (HAVCR2)*, while in LUSC we found the putative ICI *CD96-NECTIN1*. While different advanced clinical trials targeting TIGIT, including in patients affected by NSCLC, are ongoing[58], progress on TIM3 and CD96 is more limited[59]. A first-in-human phase-I study evaluating the anti-CD96 monoclonal antibody GSK6097608 as monotherapy alone or in combination with anti-PD1 (dostarlimab) started recruiting patients only recently[60]. Taken together, our data suggest that LUAD and LUSC patients might benefit from specific immunotherapy targeting ICIs as TIM3, TIGIT and CD96.

The TME plays a crucial role in modulating the population and behaviour of Mϕ[4]. We found that, compared to the adjacent non-tumour tissue, tumour resections harboured a lower proportion of monocytes but a higher proportion of monocyte-derived cells, such as mo-DC2s and anti-inflammatory Mϕ, suggestive of an enhanced monocyte differentiation in the TME[7,9]. The prevalence of anti-inflammatory Mϕ, including *STAB1* + Mϕ, exhibited an inverse relationship with the abundance of natural killer (NK) cells and T cells in the tumour environment; and the NK cells within the tumour exhibited reduced cytotoxic activity. Our results are in line with the recent findings that the removal of tumour cell debris by lung Mϕ leads to their conversion into an immunosuppressive phenotype, consequently hindering the infiltration of NK cells into the TME[27]. Mϕ with elevated levels of tumoural debris were reported to upregulate genes involved in cholesterol trafficking and lipid metabolism, a characteristic shared with anti-inflammatory Mϕ in our dataset. As a result, they downregulated co-stimulatory molecules, cytokines and chemokines[27] essential for the recruitment of CD8 + T cells, therefore becoming more immunosuppressive.

Among the Mϕ population within tumours, we also identified *STAB1* + Mϕ that exhibited the highest level of immunosuppression markers. These *STAB1* + Mϕ displayed a gene expression pattern akin to that of foetal lung Mϕ and demonstrated a modified iron metabolism, marked by the increased expression of genes associated with iron release in the TME. Therefore, we hypothesise that *STAB1* + Mϕ might play a crucial role in supporting tumour progression by sustaining the increased iron requirement of highly-cycling tumour cells[52,61]. In a subcutaneous LLC1 Lewis lung adenocarcinoma model, mice lacking *Stab1* expression in Mϕ, tumour growth was diminished. This outcome was attributed to a shift towards a pro-inflammatory phenotype in TAM and a robust infiltration of CD8 + T cells within the TME[62]. *STAB1* + Mϕ displayed a transcriptional resemblance to CAMLs, which concurrently expressed genes associated with both Mϕ and epithelial cells, and exhibited copy number alterations (CNAs) similar to those found in tumour cells. STAB1+ plays a pivotal role in facilitating the adhesion and engulfment of apoptotic cells by engaging in a specific interaction with phosphatidylserine, supporting the hypothesis of a strong interaction of a Mϕ with a tumour cell in CAMLs[63]. In previous studies, CAMLs were identified by immunofluorescence in the peripheral blood of individuals affected by various solid tumours and were proposed to facilitate the dissemination and establishment of circulating tumour cells in distant metastatic sites[16]. Here, we report their presence in multiple tumour resections, based on a combination of a compound gene expression signature, tumour-specific copy number alterations and physical proximity to tumour cells, as evident from Visium sections. Taken together, our comprehensive dataset allowed identifying a multitude of molecular changes in the Mϕ population of the lung tumour microenvironment, which will help pave the way for the development of therapeutic strategies against NSCLC.

## Methods

### Ethics and tissue acquisition

Tissue used in the research study was obtained from the Papworth Hospital Research Tissue Bank. Written consent was obtained for all tissue samples using Papworth Hospital Research Tissue Bank's ethical

approval (East of England− Cambridge East Research Ethics Committee). Human tumour and adjacent background tissues, collected from the edges of the lungs, were obtained from 25 patients following tumour resection. Human healthy lung samples were obtained from two healthy deceased donors. Both healthy samples were evaluated by an expert pathologist to exclude the presence of malignancies. The human material was provided by the Royal Papworth Tissue Bank (T02229), in accordance with the HMDMC Human Tissue Act Sample Custodian Form Version 7.0 (UK NRES REC approval reference number(s): 08/H0304/56 + 5; HMDMC 16|094). NSCLC FFPE tumour blocks ($n = 2$) used for validation of STAB1+ macrophages with Akoya were obtained from 2 different donors and purchased from BioIVT (ex-Asterand Bioscience). Informed Consent Form (ICF) and Institutional Review Board Approval Letter (IRBA) were obtained for all tissue samples.

Sex was assigned (15 male and 12 female patients/donors). Sex-based analyses were not performed due to the limited sample size. Gender was not determined.

## Tissue processing

Tissues were kept in cold complete RPMI medium (RPMI [Invitrogen] supplemented with 10% FBS [Sigma Millipore, catalogue number: F9665], 2 mM L-Glutamine [Life Technologies, catalogue number: 25030-024] and 100 U/ml Penicillin-Streptomycin [Thermofisher, catalogue number: 15140122]) until dissociation, which was performed on the same day of collection. Single-cell suspensions were generated as follows: tissues were placed into a petri dish and cut into small pieces of 2–4 mm and transferred into a 1.5-ml tube containing the digestion mix (complete RPMI media supplemented with 1 mg ml$^{-1}$ collagenase IV and 0.1 mg ml$^{-1}$ DNase I) and minced using surgical scissors. Minced tissues were incubated for 45 min at 37 °C and vortexed every 15 min. Digested tissues were passed through a 100-µm strainer into a falcon tube prefilled with cold PBS.

Cells were then centrifuged for 5 min at 300 × $g$, 4 °C and the pellet was resuspended into 1× RBC lysis buffer (eBioscience) for 2 min at room temperature, after which 20 ml of cold PBS were added to stop the lysis reaction. Cells were cryopreserved in 5% DMSO in KnockOut Serum Replacement (KOSR; Gibco™, catalogue number: 10828010) until further use.

## FACS sorting

On the day of FACS sorting, cells were rapidly thawed at 37 °C and transferred to complete RPMI media. Live-cell enrichment was performed using MACS Dead Cell Removal Kit (Miltenyi Biotec) following the manufacturer's instructions. Red blood cells were further depleted by negative selection using CD235a Microbeads (Miltenyi Biotec) and MACS LS columns (Miltenyi Biotec), following the manufacturer's instructions.

For FACS sorting, cells were stained with Zombie Aqua to exclude dead cells and the cocktail of antibodies for 30 min at 4 °C. Cells were centrifuged for 5 min at 300 × $g$, 4 °C, resuspended in 500 µl of 5% FBS in PBS and subsequently filtered into polypropylene FACS tubes.

Immune cells were sorted as live, CD45 + ; MDSC were sorted as live, CD45 + , Lineage- (Lin: CD3, CD56, CD19), CD33 + , HLA-DR-/low (Supplementary Data 22 and Supplementary Fig. 1A). Cells were sorted into a 1.5-ml tube, counted and submitted for 10x scRNA-seq library preparation.

## scRNA sequencing

Each cell suspension was submitted for 3′ single-cell RNA sequencing using Single Cell G Chip Kit, chemistry v3.1 (10x Genomics Pleasanton, CA, USA), following the manufacturer's instructions. Libraries were sequenced on an Illumina NovaSeq 6000, and mapped to the GRCh38 human reference genome using the CellRanger toolkit (version 3.1.0).

## scRNA sequencing data analysis

Integrating numerous samples, notably from diverse cancer subtypes and adjacent normal tissues, is challenging due to variations in gene programmes between samples. Consequently, these differences often hinder a coherent biological alignment when attempting simultaneous embedding. Most current integration techniques, primarily focused on batch correction, operate under the assumption of shared cell states across samples. However, while they aim to mitigate technical disparities, they might inadvertently erase genuine biological distinctions. Therefore, we applied the QC filtration and doublet removal on the merged dataset (Tumour + B/H) but we split the datasets between tumour and B/H for HVG selection, PCA, batch correction (using Harmony), clustering and annotations.

Starting from the unnormalised, uncorrected gene expression matrices produced (per sample) by the CellRanger protocol, we performed careful downstream analysis of the scRNA-seq data. For each CellRanger output (corresponding to a specific technical and biological replicate of the separate tumour, background and healthy data) we identified low-quality cells or empty droplets by applying the *barcodeRanks* and *emptyDrops* functions using the R package *DropletUtils*[64]. Following per-sample droplets removal, the complete set of cell expression matrices was merged (we merged tumour, background, and healthy samples), and quality control (QC) was applied to the resultant merged matrix. The remaining analysis is implemented using standard approaches in the *Scanpy*[65] framework. The QC is based on three parameters: the total UMI count (lower-upper threshold [400, 100,000]), the number of detected genes (lower-upper threshold [180, 6000]), and the proportion of mitochondrial gene count per cell (20% fraction upper bound). We applied *Scrublet*[66] to remove potential doublets with 0.06 as the expected doublet rate and then filtered the results using the parameter values (2 for minimum read count of cell, 3 for minimum detected cell of gene, 85 for minimum gene variability percentage, and 30 for the number of principal components used to embed the transcriptomes prior to k-nearest-neighbour graph construction). The resulting merged and filtered expression matrix is then normalised using the scaling factor 10,000, followed by log1p transformation.

For dimensionality reduction, we first selected sets of highly-variable genes (HVGs) from the initial gene set of 25,718. Starting from the HVG selection, the merged matrix was split into two separate matrices: *tumour*, and combined background/healthy which we refer to as B/H. After HVG selection, 1604 genes were selected from the tumour matrix and 1486 from B/H. From these separate HVG sets, we applied dimensionality reduction using Principal Component Analysis (PCA). Next, we performed PCA separately for tumour and for B/H and retained the top 15 components, according to the Scree plot elbow rule. The resulting matrix is then batch corrected to account for additional technical variations arising between samples which are non-biological in origin. We apply batch correction by using *harmonypy* (a Python version of the original *harmonyR*[67] package), based on recommended benchmarking[68] against other procedures.

Following between-sample batch correction, we computed a neighbourhood graph and applied Leiden[69] clustering (with Leiden resolution being 1) to the 15-dimensional *harmonised* PCA space[69]. For visualisation purposes, we used Uniform Manifold Approximation and Projection (UMAP) manifold embedding[70] to capture the global features of the 15-dimensional clustered manifold and represent the global structure in two and three dimensions. We identified top 100 representative genes for each cluster by performing the Wilcoxon signed-rank test[71] with the Bonferroni correction, followed by a filtering to obtain genes overexpressed in the target group (minimum log fold change as 0) and expressed in at least 30% of cells within the group. We did not control the fraction of gene expression of other clusters, by setting the maximum threshold as 100%. We then annotated each cell cluster according to the the expression profile of these

marker genes and the expression of other canonical genes significant for different lung cell types based on the literature (see extended results). The annotation procedure was done iteratively. With this approach we generated two separate annotated UMAPs, together with associated marker genes, for the tumour and B/H datasets.

## Contrasting cell-type abundances between different samples

To compare cell-type abundances, we calculated the proportion of each cell type within each patient and broad cell annotation in the unenriched (CD235-) samples. We contrasted cell-type proportions between groups (tumour vs. background or LUAD vs. LUSC) using a Wilcoxon rank-sum test. Finally, we corrected for multiple testing using a two-sided Bonferroni correction independently for each group analysed.

The association between the relative cell-type abundance for each immune cell type was evaluated on the Pearson's product-moment correlation coefficients.

## Label transfer

To test consistency in cell-type annotation performed separately in tumour and B/H, we performed reference-query mapping from tumour to B/H using scArches[22]. For the 828,191 immune cells (464,952 in tumour and 363,239 in B/H) identified through our separate annotations, we selected a common set of 10,000 HVGs. We first built an scVI model and trained it on the tumour dataset using broad cell types for reference, and applied scHPL method (provided in the scArches package, parameters set to use KNN classifier, 100 neighbours and with PCA dimensionality reduction) to obtain the hierarchy for the tumour cell types. We then applied the B/H dataset to the pretrained reference model for a query, and predicted B/H broad cell types based on tumour hierarchy (probability threshold set as 0.2). Finally, we compared the predicted cell types with our separate annotations in B/H using a heatmap to visualise the confusion matrix.

## CellPhoneDB

We initially identified a putative long list of cell–cell interactions differentially observed in the tumour environment by inferring statistically significant ligand–receptor pairs, and their corresponding cell types, using CellPhoneDB[29]. We treated the tumour (LUAD or LUSC), background, and healthy scRNA-seq profiles as independent datasets and ran CellPhoneDB separately. To reduce the impact of randomness in the way CellPhoneDB samples from input datasets, we required that any ligand–receptor pair of interest from the CellPhoneDB database be expressed in at least 30% of cells in a particular cell-type cluster of interest. The final ligand–receptor lists were further filtered by requiring that the mean log(1 + expression) of the ligand–receptor pair be greater than 1.0, and the Bonferroni-adjusted[72] P value be less than 0.01. From these filtered long lists, ligand–receptor pairs and corresponding cell types relevant to the tumour data are identified.

When evaluating the ligand–receptor lists calculated with CellPhoneDB, we did not run on the complete datasets due to the difficulty in scaling up the CellPhoneDB statistical permutation tests to scRNA-seq with more than $10^6$ cells. Instead, we separately stratified the tumour, healthy and background datasets such that the proportion of cell types, patients, and samples in the reduced 50% of the data recapitulated the proportions in the full dataset.

## Differential expression analysis

Differentiation expression analysis (DEA) was performed for AT2 cells, anti-inflammatory macrophages and alveolar macrophages using a pseudo-bulk approach to compare tumour versus background. Pseudobulks were built for each patient by summing raw gene counts across all cells in each cell type investigated. The patients 1 and 4 were not included in the analysis as their cancer subtype and stage were not known at the time of analysis. Since there were differences in the cell

count between datasets we downsampled the biggest cluster to the size of the smaller. The downsampling routine was repeated 100 times, such that 100 new datasets were created that match the smaller dataset. DEA was performed using sample-level pseudobulks and a Pythonic version of the *DESeq2* pipeline (py_DESeq2), including the patient information as co-variate[73]. The median adjusted p value by Benjamini−Hochberg procedure and median log2FC for each differentially expressed gene (DEG) was calculated across 100 iterations. We verified the robustness of this choice of 100 iterations by visualising the variability of the median p value across iterations, in order to assess its stability (Supplementary Fig. 6C). DEGs were filtered with median(padj)≤0.05 and |median(logFC)|≥1. Prior to performing over-representation analysis, the genes that were commonly upregulated in more than 50% of the contrasts were removed (DNAJB1, HSPA1A, HSPA1B, HSPB1, HSPE1, IGHA1, IGKC, IGLC2). DEGs were used to perform gene ontology (GO) overrepresentation using the *clusterProfiler* package[74]. To define *STAB1* + Mϕ and AMϕ gene signatures, we compared DEA results and intersected the genes significantly upregulated by *STAB1* + Mϕ (or AMϕ) compared to the other Mϕ populations in tumour.

## Trajectory inference−PAGA

To analyse myeloid cell trajectory in tumour dataset, we recomputed a neighbourhood graph from the same 15-dimensional harmonised PCA space as above, but only within myeloid cell populations. We next applied PAGA[38] within the Scanpy[65] package to the neighbourhood graph. In parallel, we computed the diffusion map and its force-directed layout for visualisation using the Pegasus package[75]. We finally overlaid the PAGA network with the diffusion map using the scVelo package. We repeated the same analysis workflow but on non-immune cells in the tumour dataset.

## Copy number analysis

We applied the CopyKAT package to the single-cell RNA-seq data to obtain copy number calls. The Copykat pipeline was extended to obtain confident copy number calls per cell, per chromosome arm, beyond the hierarchical clustering the standard pipeline produces.

Per cell copy number calls were obtained as follows: first, the regular CopyKAT (v1.0.5) pipeline was run on the unmodified UMI counts of a particular patient/environment (i.e., tumour or background) combination with default parameters, except for norm.cell.names. The norm.cell.names parameter allows for specifying which cells are used as confident diploid normals during expression normalisation. CopyKAT was set to use all cells labelled as cDC2 dendritic cells, as they are available in great numbers across all patients and an initial inspection of their expression profiles revealed no systematic copy number alterations.

After CopyKAT has completed, a calling step was applied that is aimed to call whole chromosome arm alterations in individual cells. We reasoned that, on a chromosome arm basis, the distribution of binned-and-normalised expression from CopyKAT should be significantly different (higher or lower) than the distribution of the same bins in all confidently diploid cells. For each chromosome arm, we model the distribution of all data bins from the confidently diploid cells as a normal distribution. Each bin on that same chromosome arm from a candidate aneuploid cell is then tested against that distribution. Finally, when more than 50% of bins across that chromosome arm are significant, the arm is marked as altered in that cell.

The above-described procedure yields a conservative true/false call per cell, per chromosome arm without directly distinguishing between gains and losses. To obtain a profile with gains and losses as is shown in Fig. 4A, we discretise the values for each bin in each cell: If the arm is altered and the expression value of the bin is negative: −1, if the arm is altered and the expression value is positive: +1, if the arm is unaltered: 0. The discretized values are then finally summed per bin

across all cells of a particular cell type and divided by the number of cells of that cell type to obtain the fraction of cells with an alteration as shown in Fig. 4A.

## Immunohistochemistry (IHC) and neutral lipid staining

Tissues were frozen in dry-ice-cooled isopentane and stored in air-tight tissue cryovials at −80 °C. The tissues were embedded in an optimal cutting temperature compound (OCT) and cryosectioned in a pre-cooled cryostat at 10 μm thickness on SuperFrost slides. On the day of the experiment, slides were thawed at room temperature for less than 5 min, then immersed in a fixation solution (4% PFA in PBS) for 20 min. After three washes with PBS, each section was permeabilized with freshly prepared 0.2% Triton-X100 (Sigma Aldrich) for 10 min at room temperatures, followed by three washes in PBS. Unspecific binding was blocked by incubating the sections in PBS + 2.5% BSA for 1 h at room temperature. Following two washes in PBS, sections were incubated with recombinant rabbit anti-CD68 (Abcam ab213363, 1:50) and mouse anti-STAB1 (Santa Cruz Biotechnology sc-293254, 10 μg/ml) in PBS + 0.5% BSA overnight at 4 °C. Primary antibodies were removed and sections washed three times with PBS, then incubated with the appropriate secondary antibodies (goat anti-rabbit AlexaFluor 594 and goat anti-mouse AlexaFluor 488 Abcam) 1:500 in PBS + 0.5% BSA for 2 h at room temperature, protected from light. Two confocal immunohistochemistry z-stacks each for tumour and background tissue from three patients were analysed. Using Fiji (ImageJ) software, the STAB1+ and CD68+ areas were segmented by automatic thresholding and quantified in each image of the z-stack.

To assess the levels of cholesterol and neutral lipids we further stained tumour and background tissue sections with BODIPY™ 493/503 (Invitrogen). After three washes in PBS, sections were incubated with a 10 μg/ml solution of BODIPY™ 493/503 in PBS (1:100 from a stock 1 mg/ml solution in DMSO) for 15 min at room temperature. Following four washes in PBS, sections were incubated for 90 s with TrueVIEW (Vector Laboratories), washed by immersing in PBS for 5 min, then tap-dried and mounted in VECTASHIELD Vibrance™ Antifade. Sections were imaged using a Zeiss LSM 710 confocal microscope at ×20 (Plan-Apochromat ×20/0.8 M27) and ×63 (Plan-Apochromat ×63/1.40 Oil DIC M27) magnification. Tile scans were set to cover an area of 3541 × 3542 microns for all sections. ImageJ was used to remove background BODIPY signals and calculate the area covered by the thresholded BODIPY on the stitched images. To compare the area covered by BODIPY in tumour and background, we used a paired $t$ test at a patient level, after confirming the normal distribution of the data using a Shapiro–Wilk test.

## Foetal lung integration

To investigate the oncofetal reprogramming of myeloid cells in NSCLC, we took advantage of a published scRNA-seq dataset of foetal lung myeloid cells[55] and the published "MoMac-VERSE"[[ 56]. The expression of the "STAB1 signature genes" and of the "AMϕ signature genes" across lung foetal myeloid cells was determined using the *AddModuleScore* function in Seurat v4.3. To combine foetal lung and adult lung tumour-infiltrating myeloid cells, we isolated the myeloid cells from our tumour and background datasets and integrated those with the aforementioned foetal lung myeloid dataset using the Pegasus package, following the following workflow: (i) remove rarely expressed genes (less than 10 cells), normalisation and log1p transformation, (ii) robust and highly-variable gene selection, (iii) PCA with optimal PC number determined by random matrix theory (resulting in 75 PCs), (iv) batch effect correction using Harmony[67], and (v) Leiden clustering on neighbourhood graph. The dendrogram was built by estimating the correlation distance between cell types on the harmonised PC embedding space, under complete linkage criterion of hierarchical clustering. The UMAP was computed to obtain a 2D summary of the harmonised PC space.

## 10x Genomics Visium spatial transcriptomics

Tissues were frozen in dry-ice-cooled isopentane and stored in air-tight tissue cryovials at −80 °C. Prior to undertaking any spatial transcriptomics protocol, the tissues were embedded in OCT compound and tested for RNA quality with an Agilent BioAnalyser. Tissues with RNA integrity (RIN) values > 7 were cryosectioned in a pre-cooled cryostat at 10 μm thickness. Two consecutive sections were cryosectioned at 10 μm thickness in a pre-cooled cryostat and transferred to the four 6.5 mm × 6.5 mm capture areas of the gene expression slide. Slides were fixed in methanol for 30 min prior to staining with H&E and then imaged using the Nanozoomer slide scanner. The tissues underwent permeabilization for 24 min. Reverse transcription and second strand synthesis was performed on the slide with cDNA quantification using qRT-PCR using KAPA SYBR FAST-qPCR kit (KAPA Biosystems) and analysed on the QuantStudio (ThermoFisher). Following library construction, these were quantified and pooled at 2.25 nM concentration. Pooled libraries from each slide were sequenced on NovaSeq SP (Illumina) using 150 base pair paired-end dual-indexed set-up to obtain a sequencing depth of ~50,000 reads as per 10x Genomics recommendations. The sequencing libraries were then processed by SpaceRanger (version 1.1.0) on the reference GRCh38 human reference genome to estimate gene expression on spots.

## Spatial cell typing with cell2location

We used cell2location[34] to deconvolute the cellular composition of each capture area (spot). As our scRNA-seq cells were annotated independently for tumour and the combined B/H datasets, we applied the deconvolution model separately as well, using tumour annotation to infer spatial cell composition of tumour sections, and background annotations for background datasets. Only spots with total UMI counts above 800 were used in downstream analysis.

The cell-type abundance in tumour and background sections were computed by summing up the q05 cell abundance, as estimated by cell2location, across spots that passed QC. Cell-type composition was computed by normalising each cell type's abundance with the total abundance of all cell types. We compared cell-type composition between tumour and background with Wilcoxon signed-rank test, followed by Bonferroni correction.

On tumour sections, we estimated the correlation distance on cell-type composition across valid spots, applied hierarchical clustering with complete linkage, and visualised the results as a dendrogram. In addition, we applied non-negative matrix factorisation analysis to the q05 estimation of cell-type abundance with eight factors.

## Ligand−receptor colocalization analysis

To study the expression of ligand−receptor pairs on the 10X Visium, we first binarised the expression of each gene in the LR pairs in the spots that passed QC. We considered a gene being expressed in a spot if its cell2location estimated abundance were higher than the median counts for that gene in the corresponding section. We counted spots where both genes in each LR pair were either co-expressed or not, in tumour and background sections from the same patient, and subsequently, applied the $\chi^2$ test on the contingency table. To correct for multiple comparisons, we adjusted the $P$ value using a conservative Bonferroni correction for all the LRs enriched in tumours in the cellphoneDB analysis (309 * 8 patients). LRs were considered significantly enriched in tumour if the Bonferroni-adjusted $P$ value was lower than 0.05 in at least four patients.

## Multiplexed Immunofluorescence

5 μm thick sections were generated from NSCLC FFPE tumour blocks. An antibody cocktail was prepared with optimal dilutions of each of the following conjugated antibodies: anti-human Stabilin-1 antibody (clone #840449, catalogue #MAB3825, R&D systems) was conjugated to a custom oligo barcode according to instructions in Akoya

Biosciences' antibody conjugation kit (Conjugation kit, #7000009; Akoya) while human CD68 (clone #KP1, catalogue #4550113, Akoya) and human PanCK (clone AE-1/AE-3, catalogue #4150020, Akoya) were obtained directly pre-conjugated to oligo barcodes from Akoya Biosciences. Complementary oligo-conjugated fluorophore reporters were obtained from Akoya Biosciences. Tissue multiplexed immunofluorescence staining and image acquisition were performed according to Akoya Phenocycler-Fusion user guide (PD-000011 Rev. A., Akoya). OME-TIFF files were generated and processed for image analysis.

### Image analysis

Analysis of the multiplexed immunofluorescence images (generated from Akoya Phenocycler-Fusion platform) was performed using Visiopharm (version 2023.09.3.15043 × 64) on the entire tissue area. Briefly, cell segmentation (including both nuclear and cytoplasmic segmentation) was first performed using Visiopharm's "Cell Detection, AI (Fluorescence)" (version 2023.09.3.15043 × 64) with its default parameters. After cell segmentation, Visiopharm's "Phenoplex Guided Workflow" was used. DAPI (nucleus), CD68 (cell body) and STAB1 (cell body) variables were selected and manually thresholded to define positive and negative cells for each marker and generate a co-occurrence matrix. Macrophages were defined as [DAPI + , CD68 + ] while STAB1+ macrophages were defined as [DAPI + , CD68 + , STAB1 + ].

### Reporting summary

Further information on research design is available in the Nature Portfolio Reporting Summary linked to this article.

## Data availability

The scRNA-seq and Visium datasets generated in this study are publicly available at BioStudies (https://www.ebi.ac.uk/biostudies/) with accession numbers E-MTAB-13526 and E-MTAB-13530, respectively. The remaining data are available within the Article, Supplementary Information or Source Data file Source data are provided with this paper.

## Code availability

The scripts used for all the analyses and to produce all the figures in the manuscript are available at https://gitlab.com/cvejic-group/lung and https://github.com/sdentro/copykat_pipeline.

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

## Acknowledgements

The authors are greatly thankful to the Papworth Hospital Research Tissue Bank for providing samples with data, and in particular to D. Rassl. The authors would like to thank L. Campos for the annotation of tumour histologies; A.M. Ranzoni, B. Myers and E. Panada for sample collection and processing; M. Nelson for computational support with initial clustering of scRNA-Seq and application of cell2location; Alessandro Di Tullio, GSK for insightful discussions; Cancer Research UK Cambridge Institute (CRUK CI) (Grant # CTRQQR-2021\100012) Genomics Core Facility for library preparation and sequencing services; Wellcome Sanger Institute (WSI) DNA pipelines for their contribution to sequencing the data; S. Leonard from New Pipeline Group (NPG) for pre-processing of sequencing data; the Cambridge NIHR BRC Cell Phenotyping Hub for support with cell sorting. We thank R. Möller, P. Rainer, and U. Tiemann for critically reading the manuscript. This study was conceived and funded by Open Targets (OTAR2060, A.C.); Core support grants from the Wellcome Trust and Wellcome Sanger Institute and both Wellcome and the MRC to the Wellcome Trust-Medical Research Council Cambridge Stem Cell Institute (203151/Z/16/Z, A.C.); European Research Council (CONTEXT 101043559, A.C.); Views and opinions expressed are however those of the author(s) only and do not necessarily reflect those of the European Union or the European Research Council Executive Agency. Neither the European Union nor the granting authority can be held responsible for them.

## Author contributions

A.C. conceived the study and oversaw all experiments and analysis. M.D.Z. performed experiments and analysis. H.X. led spatial transcriptomics analyses and co-analysed the scRNA-seq data. J.S.P. performed Visum experiments under O.B. supervision. Z.S. led the application of CellPhoneDB under M.G. supervision. S.C.D. led CopyCAT analysis. J.T. and S.C.A. performed Multiplexed Immunofluorescence under A.H. supervision. A.H. contributed to the interpretation of results. E.A. performed DEA. A.C. and M.D.Z. wrote the manuscript, and all authors edited and reviewed the manuscript.

## Competing interests

The authors declare no competing interests.
