## [Peer Review File · Nature Communications]

Single-cell and spatial transcriptomics analysis of non-small cell lung cancerREVIEWER COMMENTS

Reviewer #1 (Remarks to the Author): expertise in spatial transcriptomics and scRNA-seq bioinformatics

The manuscript by De Zuani et al. studies the heterogeneity of non-small cell lung cancer (NSCLC) and their microenvironment using single-cell and spatial transcriptomics, focusing on understanding the role of different populations of immune cells and their interactions in the lung cancer TME. The generated data are of interest, especially the spatial transcriptomics, and the authors do a great job of describing the main findings and their possible explanations.

However, there two main weaknesses that need to be addressed:

- I find the Introduction extremely short: The authors quickly refer to the existing literature, citing only 3 papers on single-cell analysis of lung cancer (refs 7-9) and give no context as to how their study adds to these findings. I understand that one of their main contributions is the spatial analysis, but still several results of the paper are based on scRNA-seq data alone and should be contextualized with respect to previous studies.

- Perhaps the major shortcoming is that a lot of the methodological details are not well described and the authors have not made their code or datasets available, making it impossible to fully understand or reproduce their results. The authors should ideally share figure-generating scripts for all their Figures, where the exact details of the analysis are documented. Along these lines, maybe I missed this, but it was not clear to me how the authors approached the Tumor and B/H data (merged or not). It seems straightforward and actually interesting to perform all analyses in the merged dataset and assess the similarity of different cell types (e.g., are macrophages found in tumor transcriptionally similar to the ones found in B/H). But several steps were apparently performed separately (HVG selection – why?), leaving many open questions. Few examples:
o How was PCA applied – to the merged dataset or separately for Tumor and B/H samples? The authors state “we applied dimensionality reduction using Principal Component Analysis (PCA) and retained the top 15 components for tumour and B/H”: it is not clear what they mean by “and” here.

o Same for Leiden clustering: what were the hyperparameters? I am assuming it was applied separately for T and B/H – if so, is the dotplot in Fig.1D corresponding to T or B/H? Most importantly, how reproducible are the clusters?

o UMAPs were clearly performed separately, but why? Which was the choice of hyperparameters? The authors state “and represent the global structure in two and three dimensions”: where, and most importantly why, did they perform 3D UMAPs? In Fig.1C and Fig.S1B-C, it appears that the authors have plotted the clusters one on top of the other, so that the underlying color distribution is masked. Also in Fig.S1B-C: are the three UMAPs simply repetitions? I doubt these are the 3D UMAPs the authors mentioned earlier.

- On CAMLS: The CAML cluster appears very heterogeneous: not all cells co-express these proteins, many are bimodal in EpCAM. I wonder why the CAML subclustering did not “pickup” the EpCAM-low CAMLS as a separate sub-cluster. Also, where are the CAMLS in the UMAP? I do not seem to locate any yellow cells. Also, regarding Fig 1FG, the authors state: “All clusters included cells from multiple patients, with the cluster size ranging from 2,520 to 124,459 cells”: it is actually impossible to visually assess this statement, as yellow bars are barely visible for 2-3 patients.

I found the rest of the analysis (especially the spatial analysis with cellPhoneDB) more thorough and clearly explained in the Methods and Figures. Finally, the manuscript is well written (very few minor typos, and one repetitive reference: 8 and 67).

Reviewer #2 (Remarks to the Author): expert in NSCLC immune phenotypes

In this study, single-cell RNA-seq was combined with spatial transcriptomics to unravel the differences in cellular composition and phenotypes between lung cancer tissues, adjacent non-

tumor tissue, and healthy tissue. The difference in cellular composition between tumor and adjacent non-tumor tissue was shown, with the tumors having much higher diversity compared to the adjacent lung tissue. When it comes to differences between two different tumor types, in this case LUAD and LUSC, it was shown that even though they have a similar cellular composition, they observed differences in the co-expression of several immune checkpoint inhibitors, which could pave the way for a novel therapeutic approach. Mainly, authors claim that patients might benefit from immunotherapy targeting novel immune checkpoint inhibitors such as TIM3, TIGIT, and CD96.

As the TME plays a critical role in tumor development and progression and its understanding is crucial for developing new therapies, here the authors showed increased heterogeneity of anti-inflammatory macrophages, with a newly identified subpopulation of STAB1+ macrophages abundantly present in the tumor tissue. The STAB1+ macrophages exhibited a modified iron metabolism, which could mean that they have a key role in supporting tumor progression. In conclusion, this study claims to show a multitude of molecular changes in macrophages of the TME, and that it represents the largest single-cell multiomics analysis of samples from lung cancer patients to this date. However, before making such strong claims, it would be useful to check the papers from Zilino et al. (PMID: 30979687) and Kim et al. (PMID: 32385277).

Comments

1. The cellular composition of the adjacent non-tumor tissue can vary depending on its distance from the tumor (PMID: 32699134). The authors did not specify the distance of the background tissue from the tumor, which is something that should have been taken into consideration.
2. Why were adjacent non-tumor tissue and normal lung tissue from patients with no malignancies combined for a single-cell RNA sequence? The tissue surrounding the tumor must have a completely different cellular composition than healthy tissue.
3. When identifying different cell clusters with the single-cell RNAseq in the tumors, is there any explanation why we don't see a cluster of neutrophils?
4. There is no patient data shown. One of the main aims of scRNAseq is to define and introduce new cell types with unique cellular functions. The later one can be proposed based on the correlation or association of the newly introduced cell types with patient or tumor characteristics, genomic background, and patient survival.
5. The authors claim that there are more macrophages in healthy tissue in comparison to tumor tissue. It is known that a high infiltration of macrophages is present in the tumor, and it is correlated with a poor prognosis (PMID: 33919517). Could the authors elaborate more on this, and why do they see the opposite in their data set?
6. Regarding the spatial co-localization of the L-Rs identified by cellphoneDB, the number of samples is way too low to draw any conclusions, especially when donor-dependent variations are taken into consideration.
7. The authors claim that the STAB1+ macrophages are abundantly present in the tumor tissue, which they tried to confirm with IHC staining (Figure 5F); however, to confirm this with IHC staining, there needs to be quantification done.

Reviewer #3 (Remarks to the Author): expert in NSCLC transcriptomics

De Zuani and colleagues report results of single-cell and spatial transcriptomics analysis in non-small cell lung cancer specimens. It is clear that these technologies promise to provide translational insights into biological mechanisms of these diseases. These insights are most impactful when the specimens are carefully annotated such that important characteristics that distinguish clinically relevant subtypes of disease can be correlated with biological data.

The primary concern raised in this paper is that the study design does not account for key tumor pathological criterion that will independently impact the biology of the analyzed specimens. Key examples are histological subtypes of lung adenocarcinoma (e.g. adenocarcinoma in situ, solid adenocarcinoma, mucinous adenocarcinoma subtypes are not indicated) and staging (Supplemental Table 1, column R shows Tumor stage which varies from Stage 1 to Stage 3). Additionally, there is a case without carcinoma included in the squamous cell group with pathology

labeled as dysplasia and there are 4 of 25 cases listed without histological type- this should be a rare event. Aggregating these tumors together diminishes the insights that can be derived from the detailed analyses performed.

A second study design concern is the grouping of the non-malignant specimens from both healthy subjects without cancer and from adjacent non-malignant tissue in subjects with lung cancer. It is accepted that the non-malignant lung tissue in lung cancer patients harbors a distinct molecular profile compared to healthy individuals without cancer. As presented, the study design impacts the power to derive important insights into the biology of the surrounding tumor microenvironment.

The data do show some interesting observations that have been previously reported in other publications. Confirming these prior observations with data derived from single cell and spatial transcriptomics is useful.

RESPONSE TO REVIEWERS' COMMENTS

We would like to thank the Reviewers for their useful questions and positive comments. All Reviewers acknowledged that we generated a valuable dataset with interesting observations but suggested that further clarification is needed especially in terms of the analysis that was done and grouping of the donors/patients. To address these comments we have performed additional analysis and experiments and rewrote parts of the manuscript, in particular the Introduction and Methods section.

Please find below an outline of our point-by-point response to Reviewers' comments together with additional details of the experiments and bioinformatics analysis that we did.

Reviewer #1 (Remarks to the Author): expertise in spatial transcriptomics and scRNA-seq bioinformatics

The manuscript by De Zuani et al. studies the heterogeneity of non-small cell lung cancer (NSCLC) and their microenvironment using single-cell and spatial transcriptomics, focusing on understanding the role of different populations of immune cells and their interactions in the lung cancer TME. The generated data are of interest, especially the spatial transcriptomics, and the authors do a great job of describing the main findings and their possible explanations.

However, there two main weaknesses that need to be addressed:

- I find the Introduction extremely short: The authors quickly refer to the existing literature, citing only 3 papers on single-cell analysis of lung cancer (refs 7-9) and give no context as to how their study adds to these findings. I understand that one of their main contributions is the spatial analysis, but still several results of the paper are based on scRNA-seq data alone and should be contextualized with respect to previous studies.

We would like to thank the reviewer for their suggestions related to improving the manuscript by expanding the introduction and citing more studies. We have now added more information about the previously published work while being aware of the journal's recommendations on the number of words and references in the manuscript.

- Perhaps the major shortcoming is that a lot of the methodological details are not well described and the authors have not made their code or datasets available, making it impossible to fully understand or reproduce their results. The authors should ideally share figure-generating scripts for all their Figures, where the exact details of the analysis are documented.

We have now summarised our figure-generating scripts in GitLab repository <https://gitlab.com/cvejic-group/lung>. The data has also been uploaded to ArrayExpress under accession numbers E-MTAB-13526 (scRNA-seq) and E-MTAB-13530 (Visium). In addition, we have included further details in the Methods section.

Along these lines, maybe I missed this, but it was not clear to me how the authors approached the Tumor and B/H data (merged or not). It seems straightforward and actually interesting to perform all analyses in the merged dataset and assess the similarity of different cell types (e.g., are macrophages found in tumor transcriptionally similar to the ones found in B/H). But several steps were apparently performed separately (HVG selection – why?), leaving many open questions.

Integrating numerous samples, notably from diverse cancer subtypes and adjacent normal tissues, is challenging due to variations in gene programs between samples. Consequently, these differences often hinder a coherent biological alignment when attempting simultaneous embedding. Most current integration techniques, primarily focused on batch correction, operate under the assumption of shared cell states across samples. However, while they aim to mitigate technical disparities, they might inadvertently erase genuine biological distinctions. Therefore, we applied the QC filtration and doublet removal on the merged dataset (Tumour + B/H) but we split the datasets between tumour and B/H for HVG selection, PCA, batch correction (using Harmony), clustering and annotations. Finally, we confirmed consistency in cell type annotations between T and B/H using label transfer analysis with scArches (Supp. Figure 2A).

Few examples:

o How was PCA applied – to the merged dataset or separately for Tumor and B/H samples? The authors state “we applied dimensionality reduction using Principal Component Analysis (PCA) and retained the top 15 components for tumour and B/H”: it is not clear what they mean by “and” here.

We performed PCA separately for tumour and for B/H. We have edited the text in the manuscript to improve the clarity.

The text now reads:

“From these separate HVG sets, we applied dimensionality reduction using Principal Component Analysis (PCA). Next, we performed PCA separately for tumour and for B/H and...”

o Same for Leiden clustering: what were the hyperparameters? I am assuming it was applied separately for T and B/H – if so, is the dotplot in Fig.1D corresponding to T or B/H? Most importantly, how reproducible are the clusters?

We used the Leiden algorithm as implemented in scanpy.tl.leiden, with the resolution set as 1 (default).

The reviewer is correct, the clustering was applied separately for T and B/H. The dotplot shown in Figure 1D is corresponding to the tumour. We have now added the missing information in the figure legends.

To clarify - we performed an iterative clustering procedure to identify different cell populations in the single-cell data. More precisely, we first found an initial clustering using the Leiden algorithm (with resolution = 1), next we merged clusters based on the similarity of their differentially expressed marker genes, and finally we sub-clustered broad cell populations into further refined populations. Thus, the iterative clustering allowed us to refine the initial clustering, such that initial broad clusters (e.g. mature myeloid cell clusters) containing multiple cell types (e.g. macrophages, monocytes, etc.) can be further split into lower-level cell types. Finally, we confirmed consistency in cell type annotations between T and B/H using label transfer analysis with scArches (Supp. Figure 2A).

o UMAPs were clearly performed separately, but why? Which was the choice of hyperparameters? The authors state “and represent the global structure in two and three dimensions”: where, and most importantly why, did they perform 3D UMAPs? In Fig.1C and Fig.S1B-C, it appears that the authors have plotted the clusters one on top of the other, so that the underlying color distribution is masked. Also in Fig.S1B-C: are the three UMAPs simply repetitions? I doubt these are the 3D UMAPs the authors mentioned earlier.

UMAP is done following the separate preprocessing and annotation of the two conditions (T or B/H) as we discussed above. We used Scanpy’s implementation to compute UMAP embedding coordinates, majorly with default parameters except for `n_components=3`, `init_pos='random'` and `random_state=10`. We acknowledge that 2D UMAP is more commonly used, but 3D UMAP can offer additional insights, especially when examining complex datasets. The Figure S1 shows UMAP projection of the tumour (B) and combined B/H (C) datasets after batch correction.

The reviewer is correct that the UMAPs in Figure 1C and Supp. Figure 1B, C were different because the coordinates in the embedding space were computed twice, which is not optimal. Even with the same random seed, UMAP has been reported to have a problem of reproducibility as being reported here:

<https://github.com/satijalab/seurat/issues/5514> for Seurat,
<https://github.com/scverse/scanpy/issues/2014> for Scanpy,
<https://github.com/lmcinnes/umap/issues/525> for general UMAP.

We apologise for this, and we have now computed UMAP once and re-used the same coordinates across different figures. In addition, we have corrected the way clusters are plotted so that the cell types are not on top of each other. The new UMAPs are now included in the revised manuscript.

- On CAMLS: The CAML cluster appears very heterogeneous: not all cells co-express these proteins, many are bimodal in EpCAM. I wonder why the CAML subclustering did not “pickup” the EpCAM-low CAMLs as a separate sub-cluster.

It is hard to speculate why we did not see *EPCAM*-low CAMLs as a separate subcluster, other than this feature is maybe not sufficient to drive separation of CAMLs. To explore this further, we performed differential expression analysis (DEA) between *EPCAM*-low and *EPCAM*-high CAMLs. Our analysis goes as follows: within the population of 2520 CAMLs from the tumour dataset, we first applied K-means based on *EPCAM* expression (the normalised and log_{1p} transformed value) and identified 2 sub-clusters of CAMLs, as shown in the heatmap below. Then, we performed DEA using DESeq2 between the two clusters (using raw count data pseudo-bulked by patient). Only *EPCAM* was identified with adjusted p-value below 0.05 (please see the volcano plot). This would suggest that there are no major transcriptional differences between cell populations that are *EPCAM*-low versus *EPCAM*-high.

Rebuttal Figure 1. Upper: Heatmap showing sub-clustering results of CAMLs based on *EpCAM* expression. Lower: Volcano plot showing DEGs in *EPCAM*^{low} and high subpopulations.

Also, where are the CAMLs in the UMAP? I do not seem to locate any yellow cells. Also, regarding Fig 1FG, the authors state: “All clusters included cells from multiple patients, with the cluster size ranging from 2,520 to 124,459 cells”: it is actually impossible to visually assess this statement, as yellow bars are barely visible for 2-3 patients.

We have replotted the clusters on UMAPs (as discussed above) so that CAMLs are more visible (coloured in red). We have replaced UMAPs in the main manuscript (Figure 1C) with these new ones.

Rebuttal Figure 2. UMAP projection of the tumour dataset.

We agree that it is not possible to easily visualise the contribution of each patient to each cell type in Supp. Figure 1FG. This is mainly related to the complexity of the dataset i.e. big differences in the frequency of different cell types. This information is important and for that

reason we have included the number of cells in each cell type in each patient in the Supp. Table 4.

I found the rest of the analysis (especially the spatial analysis with cellPhoneDB) more thorough and clearly explained in the Methods and Figures. Finally, the manuscript is well written (very few minor typos, and one repetitive reference: 8 and 67).

Reviewer #2 (Remarks to the Author): expert in NSCLC immune phenotypes

In this study, single-cell RNA-seq was combined with spatial transcriptomics to unravel the differences in cellular composition and phenotypes between lung cancer tissues, adjacent non-tumor tissue, and healthy tissue. The difference in cellular composition between tumor and adjacent non-tumor tissue was shown, with the tumors having much higher diversity compared to the adjacent lung tissue. When it comes to differences between two different tumor types, in this case LUAD and LUSC, it was shown that even though they have a similar cellular composition, they observed differences in the co-expression of several immune checkpoint inhibitors, which could pave the way for a novel therapeutic approach. Mainly, authors claim that patients might benefit from immunotherapy targeting novel immune checkpoint inhibitors such as TIM3, TIGIT, and CD96.

As the TME plays a critical role in tumor development and progression and its understanding is crucial for developing new therapies, here the authors showed increased heterogeneity of anti-inflammatory macrophages, with a newly identified subpopulation of STAB1+ macrophages abundantly present in the tumor tissue. The STAB1+ macrophages exhibited a modified iron metabolism, which could mean that they have a key role in supporting tumor progression. In conclusion, this study claims to show a multitude of molecular changes in macrophages of the TME, and that it represents the largest single-cell multiomics analysis of samples from lung cancer patients to this date. However, before making such strong claims, it would be useful to check the papers from Zilino et al. (PMID: 30979687) and Kim et al. (PMID: 32385277).

Comments

1. The cellular composition of the adjacent non-tumor tissue can vary depending on its distance from the tumor (PMID: 32699134). The authors did not specify the distance of the background tissue from the tumor, which is something that should have been taken into consideration.

We apologise for omitting this information in the manuscript. The pathologist dissected the adjacent background tissue from the edges of the lungs. Unfortunately, we do not have the exact distance for each sample but the background tissue is sufficiently distant from the tumour to be considered as uninvolved macroscopically normal tissue. We have now included this information in the Methods section of the manuscript (Ethics and Tissue acquisition).

2. Why were adjacent non-tumor tissue and normal lung tissue from patients with no malignancies combined for a single-cell RNA sequence? The tissue surrounding the tumor must have a completely different cellular composition than healthy tissue.

The reviewer is correct that there are differences in cell type composition between non-tumour normal background and lung from healthy deceased donors. As highlighted by the reviewer in

the previous comment, this is particularly the case for background tissue close to the tumour. The background tissues in our study were sufficiently distant from the tumour to be considered as uninvolved macroscopically normal tissue. Furthermore, the cells from the two healthy donors accounted for around 2% of cells in the B+H dataset. For those reasons, non-tumour normal background and lung from healthy deceased donors were combined for some of the analyses as discussed below. We recognise that we were not sufficiently clear about this in the manuscript, which we have now corrected.

We would like to clarify that in our analysis of cell-type abundances, (presented in Figure 1G-I), we *did not* include healthy donors. In other words, the differences in cell abundances are only reflecting the difference between the tumours and their matching backgrounds. We have now made that clearer in the text and figures.

Furthermore, in the analysis of L-R interactions we accounted for the origin of cells i.e. tumour, background and healthy tissue and calculated L-Rs in each of the datasets separately (see Supp. Figure 4D and Supp. Table 9-12 for the full list of L-Rs in each dataset). Therefore, cells from background and healthy tissue were not mixed in this analysis. Similarly, the DEA of AT2 (Figure 4D), anti-inflammatory macrophages and alveolar macrophages (Figure 5B, C) was performed in tumour *versus* background cells (i.e. no healthy cells were included).

The only analysis where healthy cells were included together with the background is the one presented in Figure 6. To address the reviewer's comment, we repeated the analysis but now combined the tumour- and background- originating myeloid cells from our dataset (without healthy cells) with myeloid and progenitor cells from a publicly available foetal lung scRNA-seq dataset. The results from this analysis confirmed the conclusions that were initially made. We have now updated the manuscript with the new figure.

Rebuttal Figure 3. Hierarchical clustering of the correlation distance calculated on each cell in the harmonised (tumour myeloid + background + foetal lung myeloid) PC space.

3. When identifying different cell clusters with the single-cell RNAseq in the tumors, is there any explanation why we don't see a cluster of neutrophils?

The reviewer correctly noted that we were not able to detect neutrophils in our dataset. Single-cell analysis of neutrophils is very challenging due to their low RNA content and high level of RNAses. More importantly, neutrophils are very sensitive to degradation after collection and in particular to the freezing-thawing cycle, and can be captured only when processing fresh samples. Samples used in this study were obtained from a tissue bank and we were not able to process them immediately after resection. Instead, single-cell suspension/tissues were frozen and subsequently defrosted on the day of processing (i.e. sorting, library preparation etc.). This resulted in neutrophils being degraded.

Also, 10x Genomics has special recommendations in terms of how to run the experiment in order to capture neutrophils: (<https://kb.10xgenomics.com/hc/en-us/articles/360004024032-Can-I-process-neutrophils-or-other-granulocytes-using-10x-Single-Cell-applications->)

We have now added this explanation in the manuscript (section - Tumours exhibit a higher diversity of immune and non-immune cells compared to adjacent lung tissue)

The revised text now reads:

“We identified clusters of myeloid cells with transcriptional signatures of monocytes, macrophages, dendritic cells (DCs), as well as mast cells, natural killer (NK) cells, T cells, B cells and non-immune cells (Figure 1C, D). We did not detect neutrophilic granulocytes, most probably due to their sensitivity to degradation after collection and in particular to the freezing-thawing cycle.”

4. There is no patient data shown. One of the main aims of scRNAseq is to define and introduce new cell types with unique cellular functions. The later one can be proposed based on the correlation or association of the newly introduced cell types with patient or tumor characteristics, genomic background, and patient survival.

We have included all available information about the patients in the Supp. Table 1. This includes age, sex, smoking history, cancer type, stage of the cancer and location. Unfortunately, we do not have information about the genomic background or patient survival. When comparing two NSCLC subtypes, we did not observe any differences in the cell type composition between LUAD and LUSC (Supp. Figure 4A). Also, when clustering patients based on the proportion of immune or non immune cells, we did not observe any association with broad tumour stage, sex or tumour type (Supplementary Figure 4B and C).

One of the interesting new cell populations that we identified in tumours and confirmed using IHC is the population of *STAB1*+ macrophages. We have now expanded our analysis using additional lung tumour samples and different spatial methodology (Akoya), and confirmed our initial findings (please see comment below and Rebuttal Figure 5). However, again we did not observe a correlation between the abundance of *STAB1*+ macrophages and the stage of cancer or cancer type.

The main difference that we observed was in the L-R interaction in LUAD *versus* LUSCs (Figure 2B-F).

5. The authors claim that there are more macrophages in healthy tissue in comparison to tumor tissue. It is known that a high infiltration of macrophages is present in the tumor, and it

is correlated with a poor prognosis (PMID: 33919517). Could the authors elaborate more on this, and why do they see the opposite in their data set?

In our analysis we did not observe a statistically significant increase in the abundance of broad macrophage population in tumour *versus* background (Figure 1H). In contrast, we saw a significant increase in the anti-inflammatory macrophages in tumour compared to background (Figure 1I). Therefore, we believe that our results are in line with previous work. We also observed a negative correlation between the frequency of anti-inflammatory macrophages and T/NK cells.

The text in the manuscript reads:

“Finally, we saw an increase in heterogeneity and proportion of anti-inflammatory M ϕ (AIM ϕ), with a subset of cycling anti-inflammatory M ϕ , *STAB1*+ M ϕ (Figure 1I) and CAMLs (Figure 1H) being abundantly present in tumour tissue. Interestingly, we found a strong negative correlation between the frequency of *STAB1*+ M ϕ /AIM ϕ and T/NK cells across patients, highlighting the key role of M ϕ in restraining the infiltration of cytotoxic cells in the lung tumour tissue (Figure 2A). This is in line with a recent work describing that monocyte-derived M ϕ in human NSCLC acquire an immunosuppressive phenotype and restrain the infiltration of NK cells”.

In addition, we saw an increase of monocytes and mo-derived DC in the tumour compared to the background, which is also in line with previously published work [1].

6. Regarding the spatial co-localization of the L-Rs identified by cellphoneDB, the number of samples is way too low to draw any conclusions, especially when donor-dependent variations are taken into consideration.

Due to the low number of tissue blocks collected from LUSC and LUAD patients, the statistical power was not sufficient to perform a comparative analysis between spatial localisation of LUAD/LUSC-specific L-Rs. We have stated that in the manuscript (page 6 of the main manuscript).

7. The authors claim that the *STAB1*+ macrophages are abundantly present in the tumor tissue, which they tried to confirm with IHC staining (Figure 5F); however, to confirm this with IHC staining, there needs to be quantification done.

We thank the reviewer for this suggestion. We have now quantified the abundance of *STAB1*+ macrophages in tumour and background tissue by calculating the percentage of *STAB1*+ signal in the CD68+ signal in each z-stack. This yields on average 30% for the tumour samples and 3% for the background ($p=0.028$, paired *t*-test). We have updated Figure 5.

Rebuttal Figure 4. Quantification of STAB1+ cells within the CD68+ macrophage population. Two confocal immunohistochemistry z-stacks each for tumour and background tissue from three patients were analysed. Using Fiji (ImageJ) software, the STAB1+ and CD68+ areas were segmented by automatic thresholding and quantified in each image of the z-stack. The fraction of the STAB1+CD68+ area is shown as a fraction of the total CD68+ area (in percentage). Error bars: standard deviation (n=3 biological replicates).

In addition, we have performed another experiment, using Akoya, on two lung cancer samples that we purchased from BioIVT. In this independent dataset we were able to confirm the presence of STAB1+ macrophages, therefore further strengthening our initial findings.

Rebuttal Figure 5. Staining for CD68, STAB1 and PanCK on tumour tissue sections using Akoya. Maximum intensity projection of Z-stacks. Inlets show a detailed magnification on a single cell. Scale bar 20 μ m.

Reviewer #3 (Remarks to the Author): expert in NSCLC transcriptomics

De Zuani and colleagues report results of single-cell and spatial transcriptomics analysis in non-small cell lung cancer specimens. It is clear that these technologies promise to provide translational insights into biological mechanisms of these diseases. These insights are most impactful when the specimens are carefully annotated such that important characteristics that distinguish clinically relevant subtypes of disease can be correlated with biological data.

The primary concern raised in this paper is that the study design does not account for key tumor pathological criterion that will independently impact the biology of the analyzed specimens. Key examples are histological subtypes of lung adenocarcinoma (e.g.

adenocarcinoma in situ, solid adenocarcinoma, mucinous adenocarcinoma subtypes are not indicated) and staging (Supplemental Table 1, column R shows Tumor stage which varies from Stage 1 to Stage 3). Additionally, there is a case without carcinoma included in the squamous cell group with pathology labeled as dysplasia and there are 4 of 25 cases listed without histological type- this should be a rare event. Aggregating these tumors together diminishes the insights that can be derived from the detailed analyses performed.

We would like to thank the reviewer for drawing our attention to the lack of sufficient information related to the type of NSCLC. The information provided in the Supp. Table 1 was not updated with the information that we received from the tissue bank, for which we apologise. We have now updated the table and provided more specific histological annotations for the 5 samples. Specifically, the sample initially defined as squamous dysplasia is actually poorly differentiated Squamous Carcinoma, and the four samples initially annotated as Non-Small Cell Lung Cancer (NSCLC) are actually 2xSquamous Cell Carcinoma, Papillary Predominant Adenocarcinoma and Adenocarcinoma (please see revised Supp. Table 1).

We grouped patients into LUSC and LUAD categories in our analysis. Further splitting of patients based on the pathological criteria within these subtypes would not yield significant insights in our study due to the lack of statistical power i.e. too low number of patients. Nevertheless, we do believe that the data provided will be of value for future studies where integration of multiple datasets across many studies and patient cohorts will allow such analysis.

A second study design concern is the grouping of the non-malignant specimens from both healthy subjects without cancer and from adjacent non-malignant tissue in subjects with lung cancer. It is accepted that the non-malignant lung tissue in lung cancer patients harbors a distinct molecular profile compared to healthy individuals without cancer. As presented, the study design impacts the power to derive important insights into the biology of the surrounding tumor microenvironment.

The reviewer is correct that there are differences in cell type composition between non-tumour normal background and lung from healthy deceased donors. This is particularly the case for background tissue close to the tumour. In our study the pathologist dissected the adjacent background tissue from the edges of the lungs. We do not have the exact distance for each sample but the background tissue is sufficiently distant from the tumour to be considered as uninvolved macroscopically normal tissue. Furthermore, the cells from the two healthy donors accounted for around 2% of cells in the B+H dataset. For those reasons, non-tumour normal background and lung from healthy deceased donors were combined for some of the analyses as discussed below. We recognise that we were not sufficiently clear about this in the manuscript, which we have now corrected.

We would like to clarify that in our analysis of cell-type abundances, (presented in Figure 1G-I), we *did not* include healthy donors. In other words, the differences in cell abundances are only reflecting the difference between the tumours and their matching backgrounds. We have now made that clearer in the text and figures.

Furthermore, in the analysis of L-R interactions we accounted for the origin of cells i.e. tumour, background and healthy tissue and calculated L-Rs in each of the datasets separately (see Supp Figure 4D and Supp. Table 9-12 for the full list of L-Rs in each dataset). Therefore, cells from background and healthy tissue were not mixed in this analysis. Similarly, the DEA of AT2 (Figure 4D), anti-inflammatory macrophages and alveolar macrophages (Figure 5B, C) was performed in tumour *versus* background cells (i.e. no healthy cells were included).

The only analysis where healthy cells were included together with the background is the one presented in Figure 6. To address the reviewer's comment, we repeated the analysis but now combined the tumour- and background- originating myeloid cells from our dataset (without healthy cells) with myeloid and progenitor cells from a publicly available foetal lung scRNA-seq dataset. The results from this analysis confirmed the conclusions that were initially made. We have now updated the manuscript with the new figure.

Rebuttal Figure 6. Hierarchical clustering of the correlation distance calculated on each cell in the harmonised (tumour myeloid + background + foetal lung myeloid) PC space.

The data do show some interesting observations that have been previously reported in other publications. Confirming these prior observations with data derived from single cell and spatial transcriptomics is useful.

References:

[1] Collin, M. and Bigley, V: Human dendritic cell subsets: an update. *Immunology* 154(1), 3-20 (2018). doi: 10.1111/imm.12888.

REVIEWER COMMENTS

Reviewer #1 (Remarks to the Author):

I would like to thank the authors for fully addressing my comments and for making the code and dataset publicly available.

Reviewer #1 (Remarks on code availability):

I have checked that the necessary files are in the given repository, the analysis appears sound but I have not re-generated all the figures myself.

Reviewer #2 (Remarks to the Author):

I would like to thank the authors for answering all the questions and providing more clarification about their work. However, there are still a few concerns I would like to point out:

1. The authors combined the healthy lung tissue from donors with no malignancies with the non-tumor tissue of lung cancer patients. They described nicely why they decided to combine these two types of tissues. However, it was not very convincing, as the non-tumor part of lung cancer patients can still be affected by inflammation; hence, I would strongly advise against combining non-tumor tissues from lung cancer patients with completely healthy lung tissues.

2. Regarding the spatial co-localization of the L-Rs, the n number is still too low to draw any conclusions, even if you do not take the statistical significance into consideration.

3. To confirm the presence of STAB1+ macrophages, the authors performed an additional experiment with Akoya. However, the n number was two, which is way too low.

Even though the paper does represent the largest single-cell multiomics analysis of samples from lung cancer patients, it is very descriptive and does not offer much novelty.

In addition, the way the manuscript is written is not very clear and concise. The authors should explain better what kind of samples are used and how many samples are used in certain experiments. In addition, experiments done with low n numbers should be omitted from the manuscript, as they do not offer conclusive results, especially when the high heterogeneity of the lung cancer patients is taken into consideration.

Reviewer #3 (Remarks to the Author):

The authors' response is sufficient to address concerns raised in the review.

RESPONSE TO REVIEWERS' COMMENTS

We would like to thank the Reviewers for their positive comments. To address the remaining questions we provided additional clarification related to methodology used in the manuscript.

Please find below an outline of our point-by-point response to Reviewers' comments.

Reviewer #2 (Remarks to the Author):

I would like to thank the authors for answering all the questions and providing more clarification about their work. However, there are still a few concerns I would like to point out:

1. The authors combined the healthy lung tissue from donors with no malignancies with the non-tumor tissue of lung cancer patients. They described nicely why they decided to combine these two types of tissues. However, it was not very convincing, as the non-tumor part of lung cancer patients can still be affected by inflammation; hence, I would strongly advise against combining non-tumor tissues from lung cancer patients with completely healthy lung tissues.

In the revised manuscript we ensured that **all analysis** presented in the manuscript (cell abundance testing, differential expression analysis, CNV analysis, L-R analysis, cell2location etc.) was performed on separate healthy and background samples, i.e., healthy and background cells **were not combined**. If the reviewer is referring to the initial integration of the healthy and background samples as presented in Figure 1 C, this was done with the purpose of ensuring consistent annotations of cell types in healthy and background samples. The annotated cell types were then separated based on their origin (background or healthy) and used separately in downstream analysis. The integration of datasets followed by clustering and annotations is a common approach.

2. Regarding the spatial co-localization of the L-Rs, the n number is still too low to draw any conclusions, even if you do not take the statistical significance into consideration.

We performed 10X Visium on two consecutive, 10 μ m sections, from eight patients, seven of which matched the samples used for the scRNA-seq. We analysed 36 sections in total (ntumour=20, nbackground=16) with an average UMI count of 6894/spot in tumour and 3350/spot in background. Several of the tumour-specific L-Rs identified by CellPhone DB colocalized significantly more in tumour than in background sections. We used appropriate statistical tests to confirm this. Therefore, we would disagree with the Reviewer that we can't draw any conclusions using this analysis.

Excerpt from Methods section:

"To study the expression of ligand-receptor pairs on the 10X Visium, we first binarised the expression of each gene in the LR pairs in the spots that passed QC. We considered a gene being expressed if its raw counts were higher than the median counts for that gene in each individual section. We counted spots where both genes in each LR pair were either co-expressed or not, in tumour and background sections from the same patient, and subsequently, used the χ^2 test on the contingency table. To correct for multiple comparisons, we adjusted the p-value using a conservative Bonferroni correction for all the LRs enriched in tumours in the cellphoneDB analysis (309 * 8 patients). LRs were considered significantly enriched in tumour if the Bonferroni-adjusted p-value was lower than 0.05 in at least four patients."

3. To confirm the presence of STAB1+ macrophages, the authors performed an additional experiment with Akoya. However, the n number was two, which is way too low. Even though the paper does represent the largest single-cell multiomics analysis of samples

from lung cancer patients, it is very descriptive and does not offer much novelty. In addition, the way the manuscript is written is not very clear and concise. The authors should explain better what kind of samples are used and how many samples are used in certain experiments. In addition, experiments done with low n numbers should be omitted from the manuscript, as they do not offer conclusive results, especially when the high heterogeneity of the lung cancer patients is taken into consideration.

In the first round of revision the Reviewer 2 commented:

“The authors claim that the STAB1+ macrophages are abundantly present in the tumor tissue, which they tried to confirm with IHC staining (Figure 5F); however, to confirm this with IHC staining, there needs to be quantification done.”

We performed quantification on N=3 tumour and N=3 background samples and reported the results, Figure 5F. In addition, we performed an experiment using Akoya Figure 5G, H and again confirmed the presence of STAB1+ macrophages. This experiment and its quantification using visiopharm have allowed us to confirm the presence of these macrophages in two different patients (N=2). We consider that these results are important to validate our data using a different approach. We believe that removing Akoya results would decrease the strength of this novel result.

Even though the paper does represent the largest single-cell multiomics analysis of samples from lung cancer patients, it is very descriptive and does not offer much novelty. In addition, the way the manuscript is written is not very clear and concise. The authors should explain better what kind of samples are used and how many samples are used in certain experiments. In addition, experiments done with low n numbers should be omitted from the manuscript, as they do not offer conclusive results, especially when the high heterogeneity of the lung cancer patients is taken into consideration.

REVIEWERS' COMMENTS

Reviewer #2 (Remarks to the Author):

The authors have effectively incorporated the reviewers' comments, suggestions, and feedback into the manuscript. Comprehensive revisions have been made to all sections, significantly enhancing the value of the work. Based on its current state, I believe the manuscript is well-suited for publication in the journal and would recommend its acceptance.